# FusionFactory: Fusing LLM Capabilities with Multi-LLM Log Data

## Abstract

The rapid advancement of large language models (LLMs) has created a diverse landscape of models, each excelling at different tasks. This diversity drives researchers to employ multiple LLMs in practice, leaving behind valuable multi-LLM log data. This naturally leads to the question of whether such logs can be fully leveraged to fuse LLMs' complementary capabilities. Although prior work has explored various strategies for integrating multiple LLMs, we argue that practical fusion must meet two essential requirements: (1) *compatibility with real-world serving scenarios* (*e.g.*, local and API-based serving), and (2) *flexibility to operate at different stages of the LLM pipeline to meet varied user needs* (*e.g.*, fine-tuning and inference stages). To this end, we introduce `LLMFusionBench`, a large-scale benchmark for LLM fusion that spans 14 tasks across five domains, with responses from 20 open-source LLMs (8B–671B) totaling 103M tokens. Building on `LLMFusionBench`, we propose `FusionFactory`, a systematic framework with three elaborated levels: **(1) query-level fusion** via tailored LLM routers, **(2) thought-level fusion** leveraging retrieved abstract reasoning templates, and **(3) model-level fusion** via distillation from top-ranked responses. Experiments show that `FusionFactory` consistently outperforms the best individual LLM across all 14 benchmarks, with the optimal fusion configuration varying across benchmarks, highlighting the promise of multi-LLM log data as a practical foundation for fusing diverse LLM capabilities.

## 1 Introduction

Large language models (LLMs), with differences in architecture design, training data, and optimization objectives, often excel at different tasks (Ahmed et al., 2024; Zhang et al., 2024b; Singhal et al., 2022; Luo et al., 2022). For example, DeepSeek-R1 achieves strong results in code generation (Guo et al., 2025), while Claude 3.7 Sonnet is particularly effective in factual retrieval and knowledge coverage (Anthropic, 2025); our work also verifies this diversity of LLM performance (Figure 2 and 3). Due to this diversity, researchers tend to utilize multiple LLMs in their workflow, notably in LLM API server platforms and agentic workflows (Zhang et al., 2024a); such usage pattern leaves us with valuable multi-LLM log data. We point out that these multi-LLM log data are a valuable vault, leading to an important research question: *can we fully leverage multi-LLM log data to fuse LLMs' complementary capabilities?*

To address this challenge, prior work has explored several different ways of integrating multiple LLMs. Some studies investigate multi-agent systems, where LLMs collaborate through division of labor (Talebirad & Nadiri, 2023; Han et al., 2024c; Chen et al., 2023b). Others develop LLM routers that dynamically assign queries to the most suitable model (Ding et al., 2024a; Dai et al., 2024b; Chen et al., 2023a; Ong et al., 2024; Feng et al., 2024b), while some existing model fusion approaches, including naive ensembles (Haber & Holtzman Gazit, 2013; Li et al., 2023b) and model merging (Yang et al., 2024b), aim to leverage multiple LLMs at the output and parameter levels, respectively. Despite their different formulations, these approaches share the same underlying goal of harnessing complementary strengths among LLMs. While these explorations mark an important step forward, fundamental challenges remain in building LLM fusion approaches that are both broadly effective and practically feasible. These challenges motivate a closer examination of what properties an effective fusion framework should possess.

In this work, we argue that effective fusion methods should ideally satisfy two key requirements. *First, effective fusion should be compatible with different LLM serving scenarios*: Real-world LLM servicing, broadly speaking, can be categorized into local serving (when model weights are avail-

Table 1: **Comparison of `FusionFactory` with existing works leveraging multi-LLM log data.** We evaluate along three fusion stages, API serving compatibility, and task domain coverage. `FusionFactory` supports all three fusion stages, is compatible with API-based serving, and covers six task domains, demonstrating broader applicability and flexibility compared to prior work.

| Method | LLM Fusion Stage | | | API Serving Compatibility | Task Domain Coverage |
|---|---|---|---|---|---|
| | Query-level | Thought-level | Model-level | | |
| RouteLLM (Ong et al., 2024) | ✓ | ✗ | ✗ | ✓ | 3 |
| GraphRouter (Feng et al., 2024b) | ✓ | ✗ | ✗ | ✓ | 4 |
| RouterBench (Hu et al., 2024) | ✓ | ✗ | ✗ | ✓ | 5 |
| LLM-Blender (Jiang et al., 2023b) | ✓ | ✗ | ✗ | ✓ | 4 |
| FuseLLM (Wan et al., 2024) | ✗ | ✗ | ✓ | ✗ | 3 |
| `FusionFactory` | ✓ | ✓ | ✓ | ✓ | **6** |

able) and API-based serving (when models are closed-source or expensive to run). While LLM distillation (Xu et al., 2024) has emerged as a popular paradigm for fusion, existing methods typically rely on logits or internal states, which are inaccessible in the generic LLM serving scenarios (*e.g.*, API-based serving). In contrast, multi-LLM log data, which is stored in text format, offers the most practical, cost-effective, and scalable foundation for studying LLM fusion, and thus forms the focus of this work. *Second, effective fusion should happen at different stages in the LLM pipeline to accommodate different users' needs*. In practice, users may or may not have the privilege of fine-tuning an LLM (due to cost and model availability considerations). Moreover, different tasks may require an LLM to fuse at different stages; for example, a RAG-based LLM system works more favorably with retrieval-based thought-level LLM fusion. In this work, we aim to systematically understand the trade-offs of LLM fusion at different stages.

Based on the above observations, we propose `LLMFusionBench`, a large-scale benchmark designed to support comprehensive studies on LLM fusion. `LLMFusionBench` provides rich supervision by including both direct and reasoning-augmented responses from multiple LLMs, as well as reusable thought templates summarized from the top-performing models. Covering 14 tasks across 6 domains, `LLMFusionBench` is built from 20 open-source LLMs ranging from 8B to 671B, making it one of the most diverse resources for studying multi-LLM integration (Table 2 and 3).

Building on `LLMFusionBench`, we introduce `FusionFactory`, a systematic framework for LLM fusion across three representative levels (comparison shown in Table 1). **(1) Query-level fusion** adapts to the varying strengths of different models and the diverse user preferences regarding token generation costs by constructing routers tailored to each query, enabling finer-grained capability fusion. Unlike traditional methods, we incorporate both direct outputs and reasoning-augmented responses (Yuan et al., 2024; Besta et al., 2024; Yao et al., 2023; Wei et al., 2022) to better address complex user queries. **(2) Thought-level fusion** improves new responses by retrieving abstract thought templates (*i.e.*, concise summaries distilled from top-performing model outputs) based on embedding similarity to past queries. This design avoids the noise of full past responses while still capturing useful reasoning patterns, which are then used as few-shot demonstrations for new queries (Song et al., 2023; Kang et al., 2023; Li et al., 2024b; Zhao et al., 2021). **(3) Model-level fusion** takes a distillation view (Sreenivas et al., 2024; Stichlmair et al., 2021), transferring complementary capabilities into a single base model by conducting supervised fine-tuning on high-quality responses aggregated across LLMs. Inspired by imitation learning in the LLM distillation domain (Hussein et al., 2017; Osa et al., 2018), for each query, we select the LLM responses with the top-k performance or the highest LLM judge scores as training data. This enables downstream deployment of a fused model without relying on multiple LLMs at inference time.

We systematically evaluate `FusionFactory` on `LLMFusionBench` and find that *fusing LLMs with `FusionFactory` yields consistent benchmark performance improvement over the best individual LLM across 14 popular LLM benchmarks*, where the optimal `FusionFactory` configuration varies across different benchmarks. Specifically, we find that: (1) *thought-level fusion achieves the best overall performance*, especially when using hybrid selection and large summaries; (2) *model-level fusion performs the worst among the three levels*, likely due to overfitting and difficulty generalizing across tasks; (3) *query-level fusion offers a good balance between performance and efficiency*, achieving decent results while requiring minimal computational overhead; (4) *gains from fusing World Knowledge and Math domains are moderate*, where these domains demand high factual accuracy or strict logical consistency, which are difficult to maintain through multi-LLM fusion. These results highlight the promise of multi-LLM log data as a practical foundation for developing fusion methods that flexibly adapt to diverse serving scenarios and fusion stages.

| Task | Query | LLM | Response | ... | Performance | Cost | LLM Judge |
|------|-------|-----|----------|-----|-------------|------|-----------|
| TriviaQA | Which Irish boxer...Cyclone? | Granite-code-direct (34b) | The Kennet, Lea,... | ... | 0.15 | 41.6 | 0.20 |
| TriviaQA | Which Irish boxer...Cyclone? | LLaMA-3-direct (70b) | The rivers Kennet,... | ... | 0.14 | 40.5 | 0.10 |
| GSM8K | Lucy has been...how much...? | LLaMA-3-ChatQA-direct (70b) | Lucy needs to buy.... | ... | 0.00 | 119.7 | 0.00 |
| GSM8K | **Lucy has been...how much...?** | **LLaMA-3-ChatQA-think (70b)** | Let me think Lucy... | ... | 1.00 | 191.7 | 0.80 |

Figure 1: **Overview of `LLMFusionBench`'s construction process.** For each task and user query, multiple LLMs generate responses in two modes: direct and reason (*i.e.*, chain-of-thoughts (CoT) (Wei et al., 2022)). These responses are then evaluated for performance, cost, and judge scores.

Table 2: **`LLMFusionBench` covers 14 tasks across 6 representative domains**.

| Domain | Tasks |
|--------|-------|
| Math | GSM8K, MATH |
| Code | MBPP, HumanEval |
| Commonsense Reasoning | CommonsenseQA, ARC, HellaSwag, OpenBookQA |
| World Knowledge | NaturalQuestions, TriviaQA |
| Reading Comprehension | SQuAD, BoolQ |
| Popular | MMLU, GPQA |

Table 3: **`LLMFusionBench` includes 20 LLMs grouped into three size-based scales.**

| Scale | LLMs |
|-------|------|
| Small | Qwen2 (7b), Mistral (7b), Qwen2.5 (7b), Gemma (7b), CodeGemma (7b), Gemma-2 (9b), LLaMA-3.1 (8b), Granite (8b), LLaMA-3 ChatQA (8b), Mistral-Nemo (12b) |
| Medium | LLaMA-3.3 Nemotron Super (49b), Granite Code (34b), LLaMA-3.1 Nemotron (51b), LLaMA-3 ChatQA (70b), LLaMA-3.1 (70b), LLaMA-3 (70b), Mixtral (8x7b) |
| Large | Palmyra Creative (122b), Mixtral (8x22b), DeepSeek-R1 (671b) |

## 2 LLMFusionBench: Benchmarking LLM Capability Fusion with Multi-LLM Log Data

In this section, we detail the construction of `LLMFusionBench` (Section 2.1) and its support for LLM capability fusion (Section 2.2), as illustrated in Figure 1.

### 2.1 LLMFusionBench Construction

As shown in Figure 1, we construct `LLMFusionBench` by sampling up to 500 training and 50 partial test queries (10:1 ratio) from each of 14 tasks across six domains (Table 2). For each query, responses from multiple LLMs are collected to support comparison and exploration of query-, thought-, and model-level fusion (Sections 4.2, 5.2, and 6.2). For the final cross-level analysis (Section 7), we also use each task's full test set to comprehensively evaluate fusion levels. Table 10 in the Appendix summarizes sample counts for training, partial test, and full test sets.

As summarized in Table 3, we select 20 LLMs across varying scales. To elicit diverse responses, we apply distinct system prompts for direct and reasoning-based outputs. Responses are evaluated with task-specific metrics (Table 7), but as shown in Figure 17, such metrics may favor short, low-information answers that hinder effective fusion. To address this, we introduce an LLM judge that scores the insightfulness of correct answers (prompt in Table 16). As shown in Table 17, the LLM judge better reflects informative value, offering greater utility for fusion.

To assess cost, we follow (Feng et al., 2024b) by estimating total token usage via GPT-2 (Radford et al., 2019) and converting it into monetary cost using Together AI's API pricing (Table 8). As shown in the right of Figure 1, collected data is stored in a DataFrame, with each row capturing task, query, LLM, response, performance, cost, and LLM judge score (details in Table 9).

### 2.2 Support for LLM Capabilities Fusion

As introduced earlier, `LLMFusionBench` captures diverse LLM responses to the same query, enabling potential for LLM capability fusion. To illustrate this, we conduct a qualitative case study using a Math query (Figure 2). We compare responses from LLaMA-3.1 Nemotron-direct (51B), LLaMA-3.1 Nemotron-think (51B), and LLaMA-3-think (70B), where -think denotes CoT prompting. These responses exhibit distinct styles: direct and concise, analytical reasoning, and detailed explanation, respectively. These variations highlight opportunities to learn diverse reasoning patterns and enhance fusion effectiveness.

| | |
|---|---|
| **\<Query>** What is the positive difference between (6² + 6²)/6 and (6² × 6²)/6? | **Response of LLaMA-3-think (70b)**: Step 1: Evaluate each expression separately. |
| **Response of LLaMA-3.1 Nemotron-direct (51b)**: 204. | Step 2: The first expression is (6² + 6²)/6, and the second expression is (6² × 6²)/6. |
| **Response of LLaMA-3.1 Nemotron-think (51b)**: To find the positive difference | Step 3: We have 6² = 36, so the expression becomes (36 + 36)/6. |
| between (6²+6²)/6 and (6²×6²)/6, we need to evaluate each expression separately. | Step 4: Adding the numbers in the numerator, we get 72/6. |
| First, let's evaluate (6²+6²)/6: (6²+6²)/6 = (36+36)/6 = 72/6 = 12 | Step 5: Dividing 72 by 6 gives us 12. |
| Next, let's evaluate (6²×6²)/6: (6²×6²)/6 = (36×36)/6 = 1296/6 = 216 | Step 6: Then we have 6² = 36, so the expression becomes (36 × 36)/6. |
| Now, we need to find the positive difference between 12 and 216: 216 - 12 = 204 | Step 7: Multiplying 36 by 36 gives us 1296, and then dividing by 6 gives us 216. |
| So, the positive difference between (6²+6²)/6 and (6²×6²)/6 is 204. | Step 8: The positive difference is 216 - 12 = 204.The final answer is: 204 |

Figure 2: **LLMs exhibit varying reasoning styles when answering the same query.** This example from the Math task compares responses from LLaMA-3.1 Nemotron-direct (51B), Nemotron-think (51B), and LLaMA-3-think (70B), where -think denotes CoT prompting. The responses show a progression from concise answers to increasingly detailed reasoning.

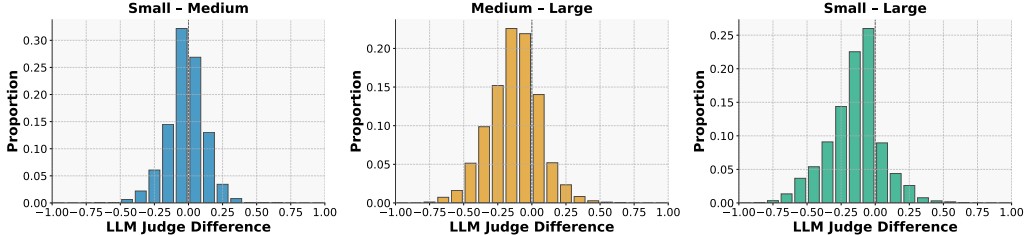

Figure 3: **LLMs of different scales show distinct strengths across queries.** For each query, we compute average LLM judge scores for small, medium, and large models (Table 3), and subtract these scores pairwise (small–medium, medium–large, small–large) to measure relative advantages. We then visualize the resulting score differences as three histograms.

To complement our analysis, we conduct quantitative evaluations using LLM judge scores. As shown in Figure 3, we compute the average judge score for each model scale per query (based on Table 3), and then calculate pairwise differences: small-medium, medium-large, and small-large. The resulting histograms reveal distinct performance gaps across scales, suggesting that models of different sizes exhibit complementary strengths and further motivating fusion via `LLMFusionBench`.

## 3 FUSIONFACTORY: FLEXIBLE FUSION OF LLM CAPABILITIES

As shown in Figure 4, we present `FusionFactory` across three representative levels of LLM capabilities fusion: query-, thought-, and model-level, corresponding to early, mid, and late fusion stages. Each level targets a distinct phase in the reasoning process. **Query-level Fusion (Early Fusion)** (Section 4) operates at the problem interpretation stage. A router analyzes the query and task characteristics to select the best LLM configuration, considering performance, cost, and LLM-based evaluations. The selected model then processes the query independently, enabling efficient task-specialized execution. **Thought-level Fusion (Mid Fusion)** (Section 5) intervenes during reasoning. An LLM summarizer distills abstract thought templates from top-k responses (ranked by performance or LLM judgment). For new queries, we retrieve similar historical queries and apply their thought templates as few-shot prompts, allowing dynamic coordination across reasoning styles. **Model-level Fusion (Late Fusion)** (Section 6) occurs after full responses are generated. A base LLM is fine-tuned using top-k high-quality outputs (selected via performance or LLM judgment) to internalize fused capabilities. This stage reflects post-hoc fusion via training rather than online collaboration. **These three fusion stages offer complementary strengths at different points in the reasoning pipeline, allowing users to flexibly choose the most suitable level of integration based on their specific serving constraints, performance goals, and resource availability.**

## 4 QUERY-LEVEL FUSIONFACTORY FUSION

### 4.1 METHODOLOGY: QUERY-LEVEL FUSION

The goal of query-level fusion is to assign the most suitable LLM for each user query, fusing LLMs' capabilities through a router. This router selects models based on the query, task info, and LLM

Figure 4: **Overview of `FusionFactory` via `LLMFusionBench` across three fusion stages:** *query-*, *thought-*, and *model-level*, corresponding to early, mid, and late fusion. At the query level, a router selects the best LLM per query using performance, cost, and judge scores. At the thought level, a summarizer extracts thought templates from top-k responses for few-shot prompting. At the model level, we fine-tune a base LLM on high-quality responses to encode fused capabilities.

features, jointly considering performance, cost, and LLM judge scores. As shown in the first panel of Figure 4, we follow Feng et al. (2024b) and define the task reward function as (arguments $(\cdot)$ omitted for brevity):

$$\text{Reward}(\cdot) = \alpha \cdot \text{Performance}(\cdot) - \beta \cdot \text{Cost}(\cdot) + \gamma \cdot \text{LLM-judge}(\cdot), \quad (1)$$

where the weights $\alpha$, $\beta$, and $\gamma$ control the relative importance of performance, cost, and the LLM judge, respectively. Base on this, for a query $q$, its corresponding task $t$, and LLM features $m$, our goal is to learn a router model $f_\phi$ such as a neural network that maximizes the Reward:

$$\phi^* = \arg\max_\phi \ \mathbb{E}_{(q,t)\sim\mathcal{D},m\sim\mathcal{M}} \left[ \text{Reward}\left(\cdot; f_\phi(q,t,m)\right) \right], \quad (2)$$

where $\mathcal{D}$ denotes the joint distribution over query-task pairs $(q,t)$ and $\mathcal{M}$ denotes the categories of different LLM features. After obtaining $\phi^*$, $f_{\phi^*}$ can select the appropriate LLM to respond to different queries and tasks.

### 4.2 EXPERIMENTAL SETTINGS: QUERY-LEVEL FUSION

**Scenario**. To explore query-level fusion, we define four routing scenarios aligned with Section 4: *Performance First*, *Balance*, *Cost First*, and *LLM Judge*. The first three examine trade-offs between performance and cost, while the last assesses routing under LLM-as-judge evaluation. These scenarios differ in their reward weights $(\alpha, \beta, \gamma)$ in Equation 1, which control the emphasis on performance, cost, and LLM-judge scores and are set to $(1.0, 0.0, 0.0)$, $(0.5, 0.5, 0.0)$, $(0.2, 0.8, 0.0)$, and $(0.0, 0.0, 1.0)$, respectively.

**Comparison Routers**. We evaluate five representative routing methods for LLM fusion: *RouterKNN* (Shnitzer et al., 2023), *RouterSVM* (Hu et al., 2024), *RouterMLP* (Shnitzer et al., 2023), *RouterBERT* (Ong et al., 2024), and *GraphRouter* (Feng et al., 2024b). We also include several static baselines: *Best LLM*, which achieves the highest performance on the training set; *LLM-Mean*, which yields the average test performance across all LLMs; and *Largest LLM*, which denotes the test performance of the largest model. Detailed baseline description can be found in Appendix C.

**Evaluation Metric**. Following (Feng et al., 2024b; Ong et al., 2024), we adopt three metrics: *Performance*, the average task accuracy over queries; *Cost*, the average inference expense of selected LLMs; and *LLMScore*, a learned evaluator's quality rating over ⟨query, ground truth, response⟩ triples. Full LLMScore details are in Appendix I.

### 4.3 RESULT ANALYSIS: QUERY-LEVEL FUSION

We report results in Table 4 with three key findings: **(1) Query-level fusion outperforms the best single LLM across all four scenarios.** As shown in Table 4, methods such as RouterMLP, RouterKNN, and RouterSVM consistently surpass the best individual LLM, with gains of at least 2% and up to 16% in performance or reward, depending on the scenario and metric. **(2) GraphRouter**

Table 4: **Multiple query-level fusion methods surpass the best single LLM across four scenarios. Bold** and underline indicate the best and second-best scores. In the Performance First, Balance, and Cost First settings, we report Performance, Cost, and Reward to evaluate how well each router balances performance and cost. The LLM Judge scenario evaluates router ability to integrate informative responses, using LLMScore and Performance.

| Scenario | Performance First | | | Balance | | | Cost First | | | LLM Judge | |
|---|---|---|---|---|---|---|---|---|---|---|---|
| Method | Performance | Cost | Reward | Performance | Cost | Reward | Performance | Cost | Reward | LLMScore | Performance |
| Best LLM | 0.546 | 0.1850 | 0.546 | 0.556 | 0.0681 | 0.244 | 0.431 | 0.0047 | 0.0823 | 0.823 | 0.431 |
| RouterKNN | 0.558 | 0.0666 | 0.558 | 0.461 | 0.0068 | 0.227 | 0.463 | 0.0067 | 0.0872 | 0.835 | 0.470 |
| RouterSVM | 0.546 | 0.0673 | 0.546 | 0.448 | 0.0084 | 0.220 | 0.442 | 0.0075 | 0.0825 | 0.790 | 0.449 |
| RouterMLP | 0.561 | 0.0652 | 0.561 | 0.482 | 0.0095 | 0.236 | 0.477 | 0.0078 | 0.0884 | 0.836 | 0.464 |
| RouterBERT | 0.528 | 0.1030 | 0.528 | 0.387 | 0.0262 | 0.180 | 0.368 | 0.0531 | 0.0310 | 0.805 | 0.414 |
| GraphRouter | 0.602 | 0.0184 | **0.602** | 0.464 | 0.0184 | **0.269** | 0.496 | 0.0047 | **0.0955** | **0.870** | **0.538** |

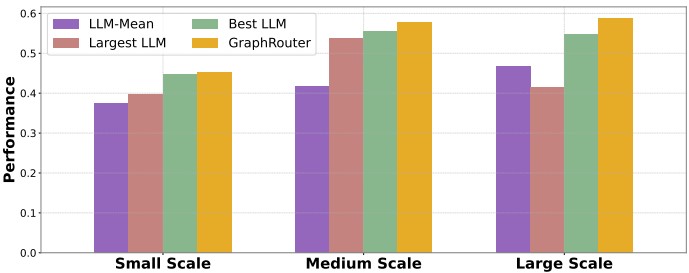

Figure 5: **GraphRouter fuses smaller LLMs to beat both top single and largest models.** Comparison across scales shows larger LLMs don't always perform best under the 1024-token constraint.

**achieves the strongest performance consistently.** It brings over 10% relative reward gains in the performance first and balance settings, 16%+ in cost first, and a 5.7% LLMScore improvement under the LLM Judge scenario. Other routers are less stable, sometimes even underperforming the Best LLM baseline, showing the challenge of effective multi-task multi-LLM fusion. **(3) GraphRouter effectively fuses smaller LLMs to outperform both the best and largest single models.** We further assess GraphRouter's ability across LLM scales (Table 3). Under the performance-first setting, it consistently beats static baselines across small, medium, and large LLMs (Figure 5), highlighting both the strength of the method and the versatility of `LLMFusionBench` in LLM fusion.

## 5 THOUGHT-LEVEL FUSIONFACTORY FUSION

### 5.1 METHODOLOGY: THOUGHT-LEVEL FUSION

At the thought level, the core insight is to summarize the responses of multiple LLMs for each query into a thought template, thereby enabling the fusion of different LLMs' capabilities to enhance few-shot responses. Specifically, as shown in the second part on the right side of Figure 4, an LLM is employed to serve as a thought summarizer $L_s$, distilling a generalized reasoning template $T_i$ from the top-k responses $R_i$ of query $q_i$ selected according to performance or LLM judge:

$$R_i = \{r_1^i, r_2^i, \ldots, r_k^i\} = \text{Top-}k\left(\{r_j^i\}_{j=1}^n \text{Score}(r_j^i)\right), \qquad T_i = L_s(R_i) \tag{3}$$

where $\{r_j^i\}_{j=1}^n$ denotes the set of responses generated by $n$ different LLMs for query $q^i$, and $\text{Score}(r_j^i)$ is given by task-specific metrics or an LLM judge.

Based on this, for new queries $q^l$, we retrieve the top-d similar queries $\mathcal{Q}^l = (q_1, q_2, ..., q_d)$ via similarity search and use their corresponding thought templates $\mathcal{T}^l = (T_1, T_2, ..., T_d)$ to enable few-shot prompting of the LLM response $R^l$:

$$\mathcal{Q}^l = \text{Top-}d\left(\mathcal{Q}, \text{sim}(q^l, q_i)\right), \qquad \mathcal{T}^l = (T_1, T_2, \ldots, T_d), \qquad R^l = \text{LLM}\left(q^l, \mathcal{T}^l\right) \tag{4}$$

### 5.2 EXPERIMENTAL SETTINGS: THOUGHT-LEVEL FUSION

**Model and Implementation**. We evaluate thought-level fusion across three model sizes and six task domains in `LLMFusionBench`. For each training query, `LLaMA-3 (70B)` generates a high-level *thought template* by summarizing the top-3 responses, selected by *Performance* and *LLMScore*, with

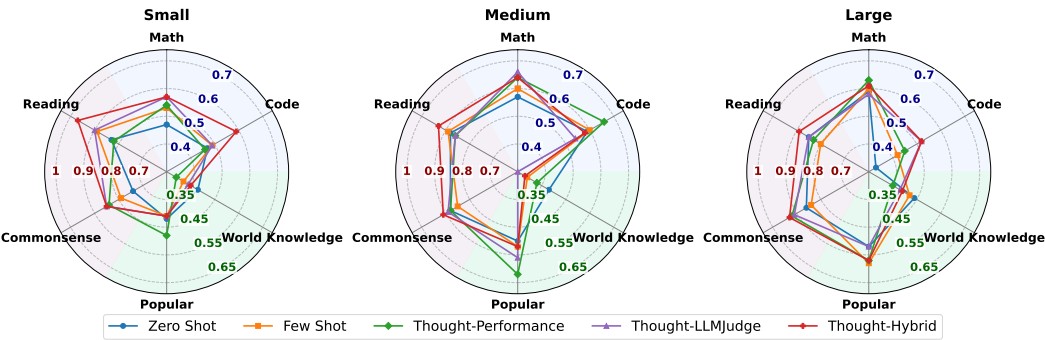

Figure 6: **Thought-level fusion improves performance across model scales and task domains.** Radar plots show accuracy per domain for small (LLaMA-3.1 8B), medium (LLaMA-3.1 Nemotron 51B), and large (Palmyra Creative 122B) models under different paradigms. The thought template approach yields the largest gains, especially on reasoning-heavy tasks like Math and Code. The hybrid strategy balances performance and response quality, often achieving the best results.

cost as tiebreaker (prompt in Table 12). To assess effectiveness, we compare three paradigms: (1) *Zero-Shot*, using only the query; (2) *Few-Shot*, retrieving three similar queries (by cosine similarity) each with three top responses; (3) *Thought Template*, using the same three similar queries, but with distilled templates. For *Thought Template*, we try three strategies: (a) *Performance-based*, using top-3 responses by *Performance*; (b) *LLMScore-based*, by *LLMScore*; (c) *Hybrid*, selecting top-5 by *Performance*, then top-3 by *LLMScore*. We experiment with *llama3-8b* (small), *llama-3.1-nemotron-51b-instruct* (medium), and *palmyra-creative-122b* (large), chosen as top performers in their scale. Accuracy is averaged by domain and shown as radar plots in Figure 6.

## 5.3 RESULT ANALYSIS: THOUGHT-LEVEL FUSION

As shown in Figure 6, thought-level fusion consistently improves performance across all model sizes. Averaged over six task domains, small, medium, and large models improve by 12.7%, 4.8%, and 8.3%, respectively, compared to zero-shot. These improvements manifest differently across reasoning types and model sizes, as detailed below. **(1) Strong gains on reasoning-intensive tasks.** Thought fusion is most effective on tasks like math and code. Hybrid fusion improves math by 21.3% (small) and 15.8% (medium); the large model (Palmyra Creative 122B) gains 57.6% in coding (from 0.33% to 0.52%). Commonsense and reading comprehension also improve (14.9% and 16.9% for small models). In contrast, performance drops on world knowledge (–7.9% for small models) and varies on popular knowledge, likely due to these tasks emphasizing factual recall over reasoning. **(2) Large models still benefit.** Thought-level fusion yields notable improvements even for large models: +57.6% in code and +6.8% in math. Medium models gain 11.9% (code) and 15.8% (math), showing the value of distilling diverse reasoning even at large model scales. **(3) Hybrid strategy offers best trade-off.** The hybrid approach balances accuracy and quality best, boosting small model accuracy by 12.7%, outperforming performance-based (3.6%) and LLMScore-based (7.3%) strategies. This balance leads to more coherent and helpful templates. See Appendix D and G for analysis of inference time and benefits of reasoning-augmented responses in thought-level fusion.

## 6 MODEL-LEVEL FUSIONFACTORY FUSION

### 6.1 METHODOLOGY: MODEL-LEVEL FUSION

The model-level aims to leverage different LLMs' high-quality responses as training data to fuse LLM capabilities. Specifically, as shown in the last part on the right side of Figure 4, we obtain the top-k responses $R_i$ of query $q_i$ selected according to performance or LLM judge as equation (3). Based on this, we can obtain a training data containing $n$ samples $Z = \{q_i, R_i\}_{i=1}^n$. Finally, we leverage $Z$ to train an LLM $f_\theta$ based on supervised fine-tuning:

$$\theta^* = \arg\min_\theta \frac{1}{n} \sum_{i=1}^n \mathcal{L}\left(f_\theta(q_i), R_i\right) \tag{5}$$

Once $\theta^*$ is obtained, the fine-tuned $f_{\theta^*}$ can generate responses for different queries.

Table 5: **Model-level fusion performance of LLaMA-3.1 (8B) on `LLMFusionBench` under different fine-tuning and selection strategies.** We compare Zero-Shot, Top-$K$ Label-only SFT, and Top-$K$ SFT across three response selection criteria: Task Performance, LLM Judge, and Hybrid. Results span six tasks, with **bold** and underline indicating the best and second-best. CS, WK, and Read denote Commonsense, World Knowledge, and Reading Comprehension.

| Sort Criterion | Task Performance | | | | | | LLM Judge | | | | | | Hybrid | | | | | |
|---|---|---|---|---|---|---|---|---|---|---|---|---|---|---|---|---|---|---|
| Scenarios | Math | Code | CS | WK | Read | Popular | Math | Code | CS | WK | Read | Popular | Math | Code | CS | WK | Read | Popular |
| Zero-Shot | 0.473 | **0.564** | 0.725 | 0.300 | 0.703 | 0.415 | _0.473_ | _0.564_ | 0.725 | **0.300** | **0.703** | 0.415 | _0.473_ | _0.564_ | 0.725 | **0.300** | 0.703 | 0.415 |
| Top-5 Label-only SFT | 0.217 | 0.510 | _0.785_ | 0.150 | 0.646 | 0.372 | 0.217 | 0.510 | 0.785 | 0.150 | 0.646 | 0.372 | 0.217 | 0.510 | 0.785 | 0.150 | 0.646 | 0.372 |
| Top-5 SFT | **0.547** | 0.503 | 0.755 | 0.290 | **0.731** | _0.468_ | 0.528 | **0.584** | 0.780 | _0.280_ | 0.662 | **0.489** | 0.443 | **0.578** | 0.790 | **0.300** | **0.728** | 0.457 |
| Top-10 Label-only SFT | 0.340 | 0.498 | **0.830** | 0.150 | 0.667 | **0.489** | 0.340 | 0.498 | **0.830** | 0.150 | _0.667_ | **0.489** | 0.340 | 0.498 | **0.830** | 0.150 | 0.667 | _0.489_ |
| Top-10 SFT | _0.509_ | _0.535_ | 0.765 | **0.310** | _0.707_ | 0.425 | 0.528 | 0.515 | _0.815_ | **0.300** | 0.647 | _0.468_ | **0.491** | 0.526 | _0.815_ | _0.250_ | _0.706_ | **0.500** |

Table 6: **Comparison across three fusion levels and six domains.** Commonsense, World, and Read denote Commonsense Reasoning, World Knowledge, and Reading Comprehension. Average indicates mean task performance. **Bold** and underline highlight the best and second-best.

| Fusion Level | Fusion LLM Size | Fusion Criteria | Math | Code | Commonsense | World | Read | Popular | Average |
|---|---|---|---|---|---|---|---|---|---|
| Zero-shot | Small | N/A | 0.602 | 0.517 | 0.715 | _0.334_ | 0.834 | 0.398 | 0.567 |
| Query | Small | Performance | 0.599 | 0.522 | 0.792 | **0.338** | 0.862 | 0.395 | 0.585 |
| Thought | Small | Hybrid-8b | _0.610_ | 0.423 | 0.778 | 0.301 | 0.956 | 0.443 | 0.585 |
| Thought | Full | Hybrid-8b | 0.544 | **0.564** | _0.799_ | 0.304 | 0.965 | _0.452_ | 0.605 |
| Thought | Small | Hybrid-70b | **0.612** | 0.550 | 0.778 | 0.312 | _0.966_ | 0.447 | _0.611_ |
| Thought | Full | Hybrid-70b | 0.588 | _0.562_ | 0.796 | 0.303 | **0.970** | **0.473** | **0.615** |
| Model | Small | LLM Judge | 0.530 | 0.510 | 0.708 | 0.286 | 0.851 | 0.410 | 0.549 |
| Model | Full | LLM Judge | 0.528 | 0.436 | **0.802** | 0.290 | 0.851 | 0.425 | 0.555 |

## 6.2 Experimental Settings: Model-level Fusion

**Model and Implementation**. To evaluate model-level fusion, we fine-tune the best-performing small LLM (LLaMA-3.1 (8B)) on `LLMFusionBench`. For each query, we rank and select LLM-generated responses using three strategies from Section 2.1: (1) **Task Performance**: top-$K$ by performance; (2) **LLM Judge**: top-$K$ by judge score; (3) **Hybrid**: first by performance, then tie-break with judge score. The selected responses from each strategy serve as training data.

Following knowledge distillation practices (Hinton et al., 2015; Xie et al., 2020), we augment training set by including each query's ground truth along with the top-$K$ LLM responses. This forms the **Top-$K$ SFT** method. We compare it with two baselines: **Zero-Shot:** directly uses LLaMA-3.1 (8B) without fine-tuning; **Top-$K$ Label-only SFT:** trains only on repeated ground truths (replicated $K + 1$ times to match training data volume). This setup tests whether model-level fusion effectively distills multi-LLM knowledge beyond supervised fine-tuning on ground truth alone.

## 6.3 Result Analysis: Model-level Fusion

We summarize results in Table 5 with three key observations: **(1) Model-level fusion yields modest but consistent gains in 4 of 6 tasks.** Compared to zero-shot baselines, fusion improves average performance under both performance- and LLM-judge–based selection, showing that fine-tuning on diverse LLM responses helps encode complementary knowledge. **(2) Fusion may hurt in domains like code due to data scarcity and task mismatch.** All fusion strategies perform worse on code, likely due to fewer samples and the divergence between code generation and QA-style reasoning in other domains, making naive multi-task training less effective. **(3) LLM Judge–based selection outperforms rigid metric-based selection.** Among the three selection strategies, _i.e._, performance, LLM Judge, and hybrid, the LLM Judge approach consistently yields the best results. This is likely because it offers finer-grained, continuous assessment compared to binary metrics (_e.g._, exact match), and better identifies informative responses.

## 7 Comparisons Across Three FusionFactory Fusion Levels

In this section, we compare the three `FusionFactory` levels on the full test set of 14 tasks (Table 6), selecting the best-performing configuration for each level: Query-level: GraphRouter using task performance as the routing criterion; Thought-level: Hybrid strategy with thought summariza-

tion; Model-level: Fine-tuning with top-$K$ responses selected by the LLM Judge. **To ensure fairness, all fusion levels are restricted to small LLMs only (LLaMA-3.1 8B) for generation, with thought/model-level also using 8B as the base model**. Table 6 further breaks down fusion settings by two factors: Fusion LLM Size (whether responses are fused from all or only small models) and Fusion Criteria (how responses are selected, *e.g.*, task performance vs. LLMScore).

Using unified experimental settings, we now summarize and compare the results across all three fusion levels. **(1) Thought-level fusion achieves the best overall performance.** Thought-Full-Hybrid-70B outperforms all settings, demonstrating its ability to integrate reasoning from diverse LLMs. However, performance gains plateau when increasing model size or response count, suggesting diminishing returns relative to cost. **(2) Model-level fusion underperforms despite access to high-quality responses.** Fine-tuned models outperform zero-shot in only 3 of 6 domains, and even fall behind on average. Likely causes include overfitting to multi-response inputs and difficulty abstracting across heterogeneous task styles during SFT. **(3) Query-level fusion achieves moderate accuracy with the lowest cost.** Despite lower accuracy, GraphRouter performs competitively while being lightweight and easy to deploy, offering a strong cost-effectiveness tradeoff. **(4) Fusion remains challenging for World Knowledge and Math domains.** In these domains, zero-shot still outperforms fusion. For World Knowledge, fusion introduces noise in tasks requiring factual precision. For Math, multi-agent reasoning disrupts sequential logic, harming accuracy.

Overall, these findings validate our central claim: multi-LLM log data, when properly structured and leveraged, enables practical and flexible fusion strategies across diverse serving needs and fusion stages. This is further demonstrated by the well-structured design of `FusionFactory` and highlights `LLMFusionBench` as a demanding benchmark for real-world multi-LLM integration.

## 8 ADDITIONAL RELATED WORK

**LLM fusion**. Recent work explores fusing multiple LLMs or reasoning traces to enhance performance, robustness, and generalization. Query-level methods (*e.g.*, LLMRouter (Li et al., 2023c), STaR (Zhou et al., 2022)) treat routing as policy learning to select the best model per query. Thought-level methods (*e.g.*, Thought Propagation (Li et al., 2023d), Tree of Thoughts (Yao et al., 2023)) aggregate reasoning paths for more consistent answers. Model-level methods (*e.g.*, FrugalGPT (Jiang et al., 2023c), Self-Consistency (Wang et al., 2022)) combine multiple model outputs or samples to approximate stronger reasoning while controlling cost. LLM-as-judge approaches (*e.g.*, LLM-as-Judge (Zhang et al., 2023), RAG Fusion (Lewis et al., 2021)) evaluate and select responses via a secondary LLM. However, most prior studies focus on a single level of fusion. We instead introduce `LLMFusionBench`, a large-scale benchmark for LLM fusion, and conduct a systematic comparison across three levels via `FusionFactory`, enabling more holistic and integrated insights.

**LLM Router**. Recent work on LLM routing focuses on improving efficiency and response quality by directing queries to the most suitable model. Early methods like RouteLLM (Ding et al., 2024b) optimize preference-aligned model selection via learned policies, while RouterBench (Hu et al., 2024) benchmarks routing decisions but does not study routing logs for capability fusion. TO-Router (Stripelis et al., 2024) and Expert Router (Anonymous, 2024) emphasize parallel dispatch for efficiency, and Glider (Li et al., 2024a) integrates global and local signals to refine expert selection. In contrast, our work focuses on capability-oriented analysis. We leverage large-scale multi-LLM log data not only for model selection, but as a form of weak supervision to systematically study and fuse LLM capabilities across three flexible levels.

## 9 CONCLUSION

In this work, we revisit LLM fusion through the lens of API-based multi-LLM log data, arguing that practical fusion must (1) be compatible with real-world serving (local and API) and (2) support integration at different pipeline stages. We introduce `LLMFusionBench`, a large-scale benchmark spanning 14 tasks and 20 LLMs with diverse response modes and reusable thought templates, and propose `FusionFactory`, a three-level framework for stage-aware fusion. Experiments show that thought-level fusion delivers the strongest gains, query-level fusion offers the best accuracy–cost trade-off, and model-level fusion lags due to generalization limits. Overall, these results validate multi-LLM log data as a practical supervision source and position `LLMFusionBench` as a demanding, real-world testbed for flexible, serving-compatible LLM fusion methods.

ETHICS STATEMENT

We adhere to the ICLR Code of Ethics. Our study uses only publicly available benchmarks and API-accessible LLMs; no human subjects, personal data, or sensitive attributes are involved. All prompts, responses, and logs are text-only and contain no personally identifiable information. We complied with the provider terms of service and licensing for all models and datasets. We acknowledge that LLM-based evaluators (LLM-as-judge) can introduce bias; to mitigate this, we report task metrics alongside judge scores and document evaluator prompts and settings.

REPRODUCIBILITY STATEMENT

We detail model configurations, prompts, training settings, and hyperparameters in Section 3, with additional implementation specifics in the Appendix C, E, H, and I. All datasets are public. We fix random seeds where applicable and specify model versions/endpoints used for API calls. Upon publication, we will release scripts for data collection (prompting and logging), training, and evaluation, as well as configuration files and thought-template prompts to enable end-to-end replication of our results.

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

## A    LIMITATIONS AND FUTURE WORK

While our work focuses on evaluating real-world task performance - particularly in domains like math and coding - future research should also investigate aspects of trustworthiness (Qi et al., 2024; Han et al., 2024b) and justice (Han et al., 2024a; Dai et al., 2024a) in the fusion process. The act of fusing multiple responses may inadvertently introduce social biases, inaccuracies, or conflicting reasoning, which could impact the reliability and ethical soundness of the LLMs. Exploring safeguards or calibration techniques to ensure more robust and equitable fusion outcomes presents a valuable direction for further study.

## B    MORE ON LLMFUSIONBENCH

Here, we list all the tasks and the metrics used for each in Table 7, and all models along with their sizes and costs in Table 8.

Table 7: **The tasks and corresponding evaluation metrics of the dataset used in constructing `LLMFusionBench`, organized by domain.**

| Domain | Tasks | Metrics |
|---|---|---|
| Math | GSM8K (Cobbe et al., 2021) | Accuracy |
| | MATH (Hendrycks et al., 2021b) | Accuracy |
| Code | MBPP (Austin et al., 2021) | Pass@1 |
| | HumanEval (Chen et al., 2021) | Pass@1 |
| Commonsense Reasoning | CommonsenseQA (Talmor et al., 2019) | Accuracy |
| | ARC (Clark et al., 2018) | Accuracy |
| | HellaSwag (Zellers et al., 2019) | Accuracy |
| | OpenBookQA (Mihaylov et al., 2018) | Accuracy |
| World Knowledge | NaturalQuestions (Kwiatkowski et al., 2019) | CEM |
| | TriviaQA (Joshi et al., 2017) | CEM |
| Reading Comprehension | SQuAD (Rajpurkar et al., 2016) | CEM |
| | BoolQ (Clark et al., 2019) | CEM |
| Popular | MMLU (Hendrycks et al., 2021a) | Multi-task Accuracy |
| | GPQA (Rein et al., 2023) | Accuracy |

## C    ROUTER BASELINE DESCRIPTION FOR QUERY-LEVEL FUSION

For the query-level fusion, we evaluate five representative routing methods to assess their LLM fusion capabilities: (1) *RouterKNN* (Shnitzer et al., 2023), a non-parametric baseline that routes by nearest neighbors in query space and selects the majority LLM label; (2) *RouterSVM* (Hu et al., 2024), a support vector machine trained on query features and task labels; (3) *RouterMLP* (Shnitzer et al., 2023), a multi-layer perceptron leveraging query embeddings and task context; (4) *RouterBERT* (Ong et al., 2024), a compact BERT classifier that encodes both query and task to predict the optimal LLM; and (5) *GraphRouter* (Feng et al., 2024b), a graph-based model that treats routing as node classification over a heterogeneous graph of queries, tasks, and LLMs with learned edge interactions. All models use `all-MiniLM-L6-v2` for embeddings, except GraphRouter, which employs Longformer (Beltagy et al., 2020) to obtain the embeddings for task/query/LLM. We also include several static baselines: *Best LLM*, which achieves the highest performance on the training set; *LLM-Mean*, which yields the average test performance across all LLMs; and *Largest LLM*, which denotes the test performance of the largest model.

Table 8: **Language Models and estimated price (in $ per 1M tokens).**

| Size Type | Model | Size | Input Price | Output Price |
|---|---|---|---|---|
| Small | Gemma-2 (9b) (Team et al., 2024b) | 9B | 0.20 | 0.20 |
| | Qwen2 (7b) (Yang et al., 2024a) | 7B | 0.20 | 0.20 |
| | Gemma (7b) (Team et al., 2024b) | 7B | 0.20 | 0.20 |
| | CodeGemma (7b) (Team et al., 2024a) | 7B | 0.20 | 0.20 |
| | Mistral (7b) (Jiang et al., 2023a) | 7B | 0.20 | 0.20 |
| | LLaMA-3.1 (8b) (Grattafiori et al., 2024) | 8B | 0.20 | 0.20 |
| | Granite (8b) (Mishra et al., 2024) | 8B | 0.20 | 0.20 |
| | LLaMA-3 ChatQA (8b) (Liu et al., 2024) | 8B | 0.20 | 0.20 |
| | Qwen2.5 (7b) (Qwen et al., 2025) | 7B | 0.20 | 0.20 |
| | Mistral-Nemo (12b) (Mistral AI, 2024) | 12B | 0.30 | 0.30 |
| Medium | Granite Code (34b) (Mishra et al., 2024) | 34B | 0.80 | 0.80 |
| | LLaMA-3.3 Nemotron Super (49b)(Wang et al., 2024) | 49B | 0.90 | 0.90 |
| | LLaMA-3.1 Nemotron (51b) (Wang et al., 2024) | 51B | 0.90 | 0.90 |
| | Mixtral (8x7b) (Jiang et al., 2024) | 56B (8×7B) | 0.60 | 0.60 |
| | LLaMA-3.1 (70b)(Grattafiori et al., 2024) | 70B | 0.90 | 0.90 |
| | LLaMA-3 ChatQA (70b) (Liu et al., 2024) | 70B | 0.90 | 0.90 |
| | LLaMA-3 (70b) (Grattafiori et al., 2024) | 70B | 0.90 | 0.90 |
| Large | Palmyra Creative (122b) (team, 2024) | 122B | 1.80 | 1.80 |
| | Mixtral (8x22b)(Jiang et al., 2024) | 176B (8×22B) | 1.20 | 1.20 |
| | DeepSeek-R1 (671b) (Guo et al., 2025) | 671B | 0.55 | 2.19 |

Table 9: **Data collection information.**

| | |
|---|---|
| **Collected Data Fields** | Task Name; Task Description; Task Description Embedding; Query; Query Embedding; Ground Truth; Metric; LLM; Input Price; Output Price; Input Tokens Num; Output Tokens Num; Performance; Cost; Response; LLM Description |

## D INFERENCE TIME FOR THOUGHT-LEVEL FUSION

**Setup and offline preprocessing.** As introduced in Table 1 and Figure 1, the thought templates used by the thought-level fusion style are summarized *offline* on LLMFusionBench. For new queries, we do not re-generate summaries; we only retrieve the most relevant thought templates and prepend them to the prompt. Retrieval is implemented with FAISS, and responses are produced using the NVIDIA NIM API. This design minimizes online overhead and keeps inference practical for API-based serving.

**Inference-time comparison.** Table 11 reports average inference times on the full test set with LLaMA-3 (8B) as the backbone across three paradigms: Zero-Shot, Few-Shot, and Thought-Hybrid. Thought-level fusion incurs only a minor increase relative to Zero-Shot and Few-Shot, which is attributable to the extra input tokens from the retrieved thought templates. The average lookup time for the thought-based method is **0.03 s**, accounting for **0.7%** of the total Thought-Hybrid inference time, which is essentially negligible. In return, the performance gains are substantial. We also expect the relative overhead to further diminish with larger base models, since total inference time increases while lookup time remains nearly constant.

## E PROMPTS FOR THOUGHT-LEVEL FUSION

Previous studies have shown that thought templates can enhance the performance of LLMs (Feng et al., 2024a; Yang et al., 2024c). However, existing templates are often either too general to a big category of questions (Yang et al., 2024c) or too specific, tailored to only one particular case (Feng et al., 2024a). Our prompt 12 is designed to strike a balance between these extremes: it is specific enough to guide similar problems effectively, yet general enough to be applicable across a range of variations.

Table 10: **Sample counts in the `LLMFusionBench` training set, partial test set, and full test set across different tasks.**

| Domain | Tasks | Train | Train Tokens | Partial Test | Test Tokens | Full Test |
|--------|-------|-------|--------------|--------------|-------------|-----------|
| Math | GSM8K (Cobbe et al., 2021) | 500 | 5.838M | 50 | 551K | 1,319 |
| | MATH (Hendrycks et al., 2021b) | 500 | 8.591M | 50 | 944K | 5,000 |
| Code | MBPP (Austin et al., 2021) | 374 | 4.702M | 50 | 623K | 500 |
| | HumanEval (Chen et al., 2021) | 120 | 2.268M | 44 | 836K | 44 |
| Commonsense | CommonsenseQA (Talmor et al., 2019) | 500 | 4.511M | 50 | 423K | 1,221 |
| Reasoning | ARC (Clark et al., 2018) | 500 | 5.387M | 50 | 489K | 1,172 |
| | HellaSwag (Zellers et al., 2019) | 500 | 7.969M | 50 | 795K | 10,042 |
| | OpenBookQA (Mihaylov et al., 2018) | 500 | 4.496M | 50 | 431K | 500 |
| World | NaturalQuestions (Kwiatkowski et al., 2019) | 500 | 4.792M | 50 | 444K | 3,610 |
| Knowledge | TriviaQA (Joshi et al., 2017) | 500 | 3.722M | 50 | 323K | 17,944 |
| Reading | SQuAD (Rajpurkar et al., 2016) | 500 | 6.572M | 50 | 646K | 10,570 |
| Comprehension | BoolQ (Clark et al., 2019) | 500 | 7.101M | 50 | 671K | 3,270 |
| Popular | MMLU (Hendrycks et al., 2021a) | 500 | 10.515M | 50 | 628K | 14,042 |
| | GPQA (Rein et al., 2023) | 400 | 10.044M | 44 | 1.2M | 44 |

Table 11: **Average inference time (seconds) on the full test set with `LLaMA-3 (8B)` as the backbone.** Thought-level fusion adds minimal overhead due to template tokens, while retrieval adds only 0.03 s on average.

| Method | Mean Inference Time (s) |
|--------|-------------------------|
| Zero-Shot | 3.74 |
| Few-Shot | 3.89 |
| Thought-Hybrid | 3.91 |

# F    DETAILED RESULTS FOR COMPARISON ACROSS THREE FUSIONFACTORY LEVELS

In this section, we present detailed results for each task across all methods from the three aforementioned fusion levels, as shown in Table 14.

# G    MORE ANALYSIS: DOES REASONING DATA HELP FOR REASONING-ORIENTED TASKS?

**Motivation.** As discussed around thought-level fusion, `LLMFusionBench` includes both reasoning and non-reasoning responses (LLM-think vs. LLM-direct).

**Analysis setup.** We analyze, under the *thought-level hybrid* selection setting, the composition of best-selected responses per task (reasoning vs. non-reasoning). This directly tests whether reasoning-style data is preferentially selected for tasks that demand more reasoning.

**Findings.** Table 15 shows that (1) reasoning ratios are higher for most reasoning-oriented tasks—**Math**, **Code**, **Commonsense Reasoning**, and **Popular**—indicating that such tasks benefit from reasoning traces. (2) In contrast, **World Knowledge** and **Reading Comprehension** skew toward non-reasoning outputs, consistent with their stronger reliance on factual recall and concise evidence extraction. Overall, these results support that reasoning-level information in `LLMFusionBench` aids selection for reasoning-heavy tasks while preserving flexibility for knowledge-intensive ones.

# H    IMPLEMENTATION DETAILS OF FUSIONFACTORY

We utilize NVIDIA API[1] for our API calling in data generation and model inference. We also use a NVIDIA A6000 GPU to obtain all the embeddings for queries/tasks/LLMs described in section 4.2.

---

[1] https://build.nvidia.com/nim

Table 12: **Prompts for thought template creation**.

---

Given this question and example solutions, extract a concise thought template that captures the effective reasoning pattern and can serve as guidance:

**Question:** [QUERY]

**Here are 3 high-performing solutions:**
Solution 1: [Response 1]
Solution 2: [Response 2]
Solution 3: [Response 3]

**Please create a concise and clear thought template (1–5 sentences total) focusing on:**

- **Core Task Summarization:** Identify the core problem type and general approach needed (1 sentence).
- **Reasoning Step:** Provide a clear chain of thought to address this problem (1–3 sentences).
- **Answer Template:** Describe the preferred answer format or structure (1 sentence).

**Your template should be specific enough to guide similar problems but general enough to work across variations.**

**Thought Template:**

---

Table 13: **The model names and their specific settings.** We introduce these model names in Table 14 in terms of the aspects corresponding to Table 6, namely Fusion Level, Fusion LLM Size, and Fusion Criteria.

| Model Name | Fusion Level | Fusion LLM Size | Fusion Criteria |
|---|---|---|---|
| **Zero-shot** | Zero-shot | Small | N/A |
| **Query-Small** | Query | Small | Performance |
| **Thought-Small-8b** | Thought | Small | Hybrid-8b |
| **Thought-Full-8b** | Thought | Full | Hybrid-8B |
| **Thought-Small-70b** | Thought | Small | Hybrid-70B |
| **Thought-Full-70b** | Thought | Full | Hybrid-70b |
| **Model-Small** | Model | Small | LLM Judge |
| **Model-Full** | Model | Full | LLM Judge |

For model-level fusion, we adopt LLaMA-Factory[2] to fine-tune the model with LoRA, which is conducted on NVIDIA A6000 GPUs. For LoRA, we apply adaptation to all transformer layers with a rank of 8. Inputs are processed with a maximum context length of 2048 tokens unless otherwise specified. The training uses a per-device batch size of 8 and a gradient accumulation step of 4, resulting in an effective batch size of 32. We set the learning rate to 1e-4 and train the model for 3 full epochs. A cosine learning rate scheduler is used with a warmup ratio of 0.1 to ensure stable convergence. Training is conducted using bfloat16 precision, and Flash Attention 2 is enabled to accelerate attention computation. Note that, as shown in Table 10, the training data in the Code domain is relatively smaller and more challenging compared to other domains. Therefore, following the approach adopted in previous work (Roziere et al., 2023; Li et al., 2023a), we train a separate model for the Code domain, while a unified model is trained for the remaining domains.

---

[2]https://github.com/hiyouga/LLaMA-Factory

Table 14: **Detailed results for all levels of fusion across all benchmarks.**

| Scenario | Task | Zero-shot | Query-Small | Thought-Small-8b | Thought-Full-8b | Thought-Small-70b | Thought-Full-70b | Model-Small | Model-Full |
|---|---|---|---|---|---|---|---|---|---|
| Math | GSM8K | 0.749 | 0.7983 | 0.771 | 0.651 | 0.771 | 0.749 | 0.715 | 0.742 |
| | MATH | 0.455 | 0.4000 | 0.449 | 0.437 | 0.452 | 0.427 | 0.344 | 0.314 |
| Code | HumanEval | 0.432 | 0.4545 | 0.546 | 0.530 | 0.500 | 0.523 | 0.566 | 0.485 |
| | MBPP | 0.601 | 0.5900 | 0.300 | 0.598 | 0.600 | 0.600 | 0.455 | 0.386 |
| Commonsense Reasoning | ARC (Challenge) | 0.736 | 0.8276 | 0.819 | 0.839 | 0.819 | 0.835 | 0.689 | 0.812 |
| | CommonsenseQA | 0.723 | 0.7371 | 0.757 | 0.770 | 0.757 | 0.766 | 0.716 | 0.792 |
| | HellaSwag | 0.649 | 0.7424 | 0.745 | 0.757 | 0.745 | 0.750 | 0.727 | 0.794 |
| | OpenBookQA | 0.750 | 0.8620 | 0.792 | 0.830 | 0.792 | 0.834 | 0.700 | 0.810 |
| World Knowledge | NaturalQuestions | 0.473 | 0.4716 | 0.409 | 0.414 | 0.427 | 0.416 | 0.404 | 0.414 |
| | TriviaQA | 0.195 | 0.2034 | 0.192 | 0.193 | 0.197 | 0.189 | 0.168 | 0.165 |
| Reading Comprehension | SQuAD | 0.825 | 0.8746 | 0.921 | 0.947 | 0.948 | 0.950 | 0.849 | 0.863 |
| | BoolQ | 0.843 | 0.8492 | 0.991 | 0.983 | 0.985 | 0.990 | 0.852 | 0.838 |
| Popular | GPQA | 0.186 | 0.1818 | 0.273 | 0.227 | 0.273 | 0.273 | 0.205 | 0.182 |
| | MMLU | 0.610 | 0.6073 | 0.612 | 0.676 | 0.621 | 0.672 | 0.614 | 0.668 |

Table 15: **Reasoning vs. Non-Reasoning ratios under thought-level hybrid selection (by domain).**

| Task Domain | Reasoning Ratio (%) | Non-Reasoning Ratio (%) |
|---|---|---|
| Math | 51.25 | 48.75 |
| Code | 53.18 | 46.82 |
| Commonsense Reasoning | 71.00 | 29.00 |
| World Knowledge | 37.90 | 62.10 |
| Reading Comprehension | 44.10 | 55.90 |
| Popular | 57.50 | 42.50 |

## I  PROMPTS FOR LLM-AS-JUDGE SCORE GENERATION

We present the prompt for LLM-As-Judge Score Generation in Table 16.

Table 16: **Prompts for LLM-As-Judge score generation.**

---

You are an expert judge evaluating the quality of an AI model's response. Please score the response based on the following criteria:

1. **Correctness (0–1):** Is the answer correct according to the ground truth?

2. **Thought Process (0–1):** Does the response show clear reasoning and explanation?

3. **Training Data Quality (0–1):** Is the response well-structured and suitable for supervised fine-tuning?

**Query:** {query}

**Ground Truth:** {ground_truth}

**Response:** {response}

Please provide a single score from 0 to 1, where:

- 0: Incorrect answer
- 0.3: Correct answer but minimal thought process
- 0.6: Correct answer with some thought process
- 0.8: Correct answer with good thought process
- 1.0: Correct answer with excellent thought process and well-suited for training

Return the score in the following format:

```
<answer>SCORE</answer>
```

---

Table 17: **Comparison of performance and LLM judge for responses.** We selected responses from two different LLMs on the Math and GSM8K tasks, and obtained their respective Performance and LLM Score. It can be observed that while Performance effectively reflects the correctness of responses, it fails to distinguish the amount of information provided in responses that are all correct. In contrast, LLM Score compensates for this limitation by capturing the informational richness of responses, thereby offering greater potential for enhancing LLM capabilities fusion.

| Question of Math | |
|---|---|
| Simplify $(3p^3 - 5p + 6) + (4 - 6p^2 + 2p)$. Express your answer in the form $Ap^3 + Bp^2 + Cp + D$, where $A$, $B$, $C$, and $D$ are numbers (possibly negative). | |
| **Answer of Gemma-2 (9b)** | **Answer of Gemma-2-think (9b)** |
| $\boxed{3p^3 - 6p^2 - 3p + 10}$ | Here's how to simplify the expression: **1. Remove the parentheses:** Since we are adding, the parentheses don't affect the order of operations. $3p^3 - 5p + 6 + 4 - 6p^2 + 2p$ **2. Combine like terms:** Identify terms with the same variable and exponent, and combine their coefficients. $3p^3 - 6p^2 + (-5p + 2p) + (6 + 4)$ **3. Simplify:** $3p^3 - 6p^2 - 3p + 10$ $\boxed{3p^3 - 6p^2 - 3p + 10}$ |
| **Performance**: 1.0 **LLMScore**: 0.3 *Correct answer but minimal thought process* | **Performance**: 1.0 **LLMScore**: 0.8 *Correct answer with good thought process* |
| **Question of GSM8K** | |
| Mr. Maximilian has a rental building that he collects rent from every month. The number of units in the building is 100. If the building is 3/4 occupied for a whole year, and each resident of the building pays a rent of $400, calculate the amount of money Mr. Maximilian receives in that year. | |
| **Answer of Qwen2.5 (7b)** | **Answer of Qwen2.5-think (7b)** |
| To calculate the amount of money Mr. Maximilian receives in a year, we can follow these steps: **1. Calculate the number of occupied units:** - The building has 100 units. - It is 3/4 occupied. - Number of occupied units = $(100 \times \frac{3}{4} = 75)$. **2. Calculate the monthly rent collected:** - Each resident pays $400 per month. - Monthly rent collected = $(75 \times 400 = 30,000)$ dollars. **3. Calculate the annual rent collected:** - There are 12 months in a year. - Annual rent collected = $(30,000 \times 12 = 360,000)$ dollars. Therefore, Mr. Maximilian receives $360,000 in rent for the year. | Sure, let's break down the problem step by step to find out how much money Mr. Maximilian receives in a year from the rent. **Step 1: Calculate the Number of Occupied Units** 1. The total number of units in the building is 100. 2. The building is 3/4 (or 75%) occupied. To find the number of occupied units: Number of occupied units = $\frac{3}{4} \times 100 = 75$ **Step 2: Calculate the Monthly Rent Collection** 1. Each resident pays a rent of $400 per month. 2. There are 75 occupied units. To find the total monthly rent collection: Monthly rent collection = $75 \times 400 = 30,000$ **Step 3: Calculate the Annual Rent Collection** 1. The monthly rent collection is $30,000. 2. There are 12 months in a year. To find the total annual rent collection: Annual rent collection = $30,000 \times 12 = 360,000$ **Final Answer** Mr. Maximilian receives $360,000 in rent for the year. |
| **Performance Score**: 1.0 **LLMScore**: 0.9 *Correct answer with very good thought process* | **Performance Score**: 1.0 **LLMScore**: 1.0 *Correct answer with excellent thought process* |

## J    DESCRIPTIONS FOR TASKS AND LLMS

Using task and model descriptions generated by LLMs enhances the expressiveness and generalization capability of the router. In this work, we present descriptions of various tasks and LLMs generated by GPT-4o. Specifically, GPT-4o captures the unique characteristics and challenges associated with each task, as well as the size, cost, and distinct strengths of different LLMs. These detailed descriptions are provided in the tables below.

## K    CASE STUDY FOR ALL LEVELS

To better understand the behavior of LLMs under different fusion strategies, we conduct a series of case studies spanning diverse task types, including Math, Code, Commonsense Reasoning, World Knowledge, Reading Comprehension, and Popular questions. Each example is evaluated under four levels: Zero-shot, Query-level, Thought-level, and Model-level. These comparative analyses provide qualitative insights into how reasoning granularity and intervention level impact model performance, faithfulness, and error patterns. See Tables 18, 19, 20, 21, 22, and 23 for detailed comparisons across tasks.

Table 18: **Case study - Math.** An example math question used to compare model responses across different settings. This case probes multi-step numerical reasoning, arithmetic consistency, and symbolic manipulation under zero-shot, query-level, thought-level, and model-level settings.

| Question of Math | | | |
|---|---|---|---|
| Janet's ducks lay 16 eggs per day. She eats three for breakfast every morning and bakes muffins for her friends every day with four. She sells the remainder at the farmers' market daily for $2 per fresh duck egg. How much in dollars does she make every day at the farmers' market? | | | |
| **Zero-shot** | **Query-level** | **Thought-level** | **Model-level** |
| To find out how much Janet makes every day at the farmers' market, we need to first determine how many eggs she has left after eating and baking. Janet's ducks lay 16 eggs per day. She eats 3 for breakfast and bakes 4 for muffins, so she uses 3 + 4 = 7 eggs. To find out how many eggs she has left, we subtract the number of eggs she uses from the total number of eggs laid: 16 - 7 = 9 eggs. Janet sells the remaining 9 eggs at the farmers' market for $2 per egg. To find out how much she makes, we multiply the number of eggs she sells by the price per egg: 9 x $2 = $18. Therefore, Janet makes $18 every day at the farmers' market. | To find out how much Janet makes every day at the farmers' market, we need to follow these steps: 1. Calculate the total number of eggs laid by the ducks per day. 2. Calculate the number of eggs Janet eats for breakfast and bakes for her friends. 3. Calculate the number of eggs left over after Janet eats and bakes. 4. Calculate the amount of money Janet makes from selling the leftover eggs. Step 1: Calculate the total number of eggs laid by the ducks per day. Janet's ducks lay 16 eggs per day. Step 2: Calculate the number of eggs Janet eats for breakfast and bakes for her friends. Janet eats 3 eggs for breakfast and bakes 4 eggs for her friends, so she uses a total of 3 + 4 = 7 eggs. Step 3: Calculate the number of eggs left over after Janet eats and bakes. The total number of eggs laid is 16, and Janet uses 7 eggs, so the number of eggs left over is 16 - 7 = 9. Step 4: Calculate the amount of money Janet makes from selling the leftover eggs. Janet sells the leftover eggs for $2 each, and she has 9 eggs left over, so she makes 9 x $2 = $18. Therefore, Janet makes $18 every day at the farmers' market. `<answer>$18</answer>` | To solve this problem, I will use the thought template for calculating the remaining quantity of items after filling containers with a specific capacity. **Core Task Summarization**: Determine the daily earnings from selling eggs at the farmers' market. **Reasoning Step**: 1. Calculate the total number of eggs laid by the ducks per day: 16 eggs/day 2. Calculate the total number of eggs eaten by Janet for breakfast: 3 eggs/day 3. Calculate the total number of eggs used for baking muffins: 4 eggs/day 4. Calculate the total number of eggs used for breakfast and baking: 3 + 4 = 7 eggs/day 5. Calculate the remaining number of eggs to be sold at the farmers' market: 16 - 7 = 9 eggs/day 6. Calculate the daily earnings from selling eggs at the farmers' market: 9 eggs/day * $2/egg = $18/day **Answer Template**: The final answer is a numerical value representing the daily earnings from selling eggs at the farmers' market. The final answer is: $18 | To calculate how much Janet makes every day at the farmers' market, we need to determine how many eggs she has left after using some for breakfast and baking muffins. Then, we can calculate the remaining eggs and multiply that by the price per egg. 1. **Calculate the total eggs laid per day:** Janet's ducks lay 16 eggs per day. 2. **Calculate the eggs used for breakfast:** She eats three eggs for breakfast every morning. 3. **Calculate the eggs used for baking muffins:** She bakes muffins with four eggs every day. 4. **Calculate the total eggs used:** 3 (breakfast) + 4 (muffins) = 7 eggs. 5. **Calculate the remaining eggs:** 16 (total eggs) - 7 (eggs used) = 9 eggs. 6. **Calculate the money made at the farmers' market:** 9 eggs * $2 per egg = $18. So, Janet makes $18 every day at the farmers' market. |

## L    THE USE OF LARGE LANGUAGE MODELS (LLMS)

We used large language models (LLMs) solely for minor editing purposes, such as checking grammar and improving the readability of the text.

Table 19: **Case study - Code.** A representative code generation problem requiring functional understanding and list manipulation in Python. The table shows how different levels affect the model's ability to parse task requirements, implement correct logic, and produce syntactically valid code.

| Question of Code | | | |
|---|---|---|---|
| You are an expert Python programmer. Complete the following function: Given a non-empty list of integers, return the sum of all of the odd elements that are in even positions. Examples: solution([5, 8, 7, 1]) ==¿ 12, solution([3, 3, 3, 3, 3]) ==¿ 9, solution([30, 13, 24, 321]) ==¿ 0 | | | |
| **Zero-shot** | **Query-level** | **Thought-level** | **Model-level** |
| [BEGIN] def solution(lst): return sum(num for i, num in enumerate(lst) if i % 2 == 0 and num % 2 != 0) [Done] | [BEGIN] To solve this problem, we need to iterate through the given list and check two conditions for each element: 1. The element is at an even position (0-indexed, so positions 0, 2, 4, ... are even). 2. The element is odd. If both conditions are met, we add the element to our running total. Here's how we can implement this in Python: ```python def solution(lst): total = 0 for i in range(0, len(lst), 2): # Iterate through even positions if lst[i] % 2 != 0: # Check if the element is odd total += lst[i] # Add the element to the total return total ``` Let's test this function with the provided examples: ```python print(solution([5, 8, 7, 1])) # Output: 12 (5 + 7) print(solution([3, 3, 3, 3, 3])) # Output: 9 (3 + 3 + 3) print(solution([30, 13, 24, 321])) # Output: 0 (no odd elements at even positions) ``` [Done] <answer>total += lst[i] if lst[i] % 2 != 0 and i % 2 == 0</answer> | [BEGIN] def solution(lst): return sum(num for i, num in enumerate(lst) if i % 2 == 0 and num % 2 != 0) [Done] This solution uses a generator expression to iterate over the list, checking each element's position (i) and value (num). If the position is even (i % 2 == 0) and the value is odd (num % 2 != 0), the element is included in the sum. The 'enumerate' function is used to get both the index and value of each element in the list. | [BEGIN] result = 0; for i in range(len(lst)): if lst[i] % 2 != 0 and (i + 1) % 2 == 0: result += lst[i]; return result [Done] |

Table 20: **Case study - Commonsense Reasoning.** A question testing commonsense reasoning in everyday scenarios. This case highlights how different settings influence the model's ability to eliminate distractors, rely on implicit knowledge, and select the most plausible answer.

| Question of Commonsense Reasoning | | | |
|---|---|---|---|
| A revolving door is convenient for two direction travel, but it also serves as a security measure at a what? (A) bank (B) library (C) department store (D) mall (E) new york | | | |
| **Zero-shot** | **Query-level** | **Thought-level** | **Model-level** |
| (C) department store | Let's think through this step by step: 1. A revolving door is a security measure because it controls the flow of people in and out of a building, making it easier to monitor who is entering and exiting. 2. Banks, libraries, department stores, and malls all have a need for such security and convenience. 3. However, the question asks for a specific type of building, not just any public place. 4. Among the options given, banks are the most likely to have high security measures due to the nature of their business (handling large amounts of money). 5. Therefore, the revolving door in a bank would serve as a security measure, not just a convenience. Final answer: <answer>(A) bank</answer> | To solve this question, I will use the thought templates provided. **Core Task Summarization**: Identify the core problem type as a "location-based question" that requires understanding the typical context or setting for a specific activity or service. **Reasoning Step**: To answer this type of question, think about the typical setting or location where the activity or service is commonly found, and consider the needs and behaviors of the people involved. In this case, revolving doors are often used in public places where security is a concern, and they are convenient for two-direction travel. **Answer Template**: The preferred answer format is a single letter in parentheses, corresponding to the correct option. Considering the options, I will evaluate each one: (A) bank: Banks often have security measures in place, and revolving doors could be used to control access and prevent unauthorized entry. (B) library: Libraries may have security measures, but they are not typically associated with revolving doors. (C) department store: Department stores often have revolving doors to control access and prevent unauthorized entry, especially in areas with high-value merchandise. (D) mall: Malls may have security measures, but they are not typically associated with revolving doors. (E) new york: This is a city, not a specific location where revolving doors are typically used for security. Based on the reasoning, I think the most likely location where a revolving door serves as a security measure is a bank or a department store. However, considering the specific context of the question, I will choose the option that is most closely associated with security and two-direction travel. Therefore, my final answer is: (A) | Here's the answer: (C) department store **Explanation:** Revolving doors are often used in department stores to control access and maintain a comfortable indoor climate while also allowing for two-directional travel. They are commonly found at entrances to such establishments. **Let me know if you'd like me to clarify anything!** (I'll be happy to provide more context if needed.) |

Table 21: **Case study - World Knowledge.** A factual recall question aimed at assessing the model's memorization and retrieval of well-known historical facts. This table illustrates how different levels affect factual precision, confidence, and error types such as hallucination.

| Question of World Knowledge | | | |
|---|---|---|---|
| Who got the first Nobel Prize in Physics? | | | |
| **Zero-shot** | **Query-level** | **Thought-level** | **Model-level** |
| The first Nobel Prize in Physics was awarded in 1901 to Wilhelm Conrad Röntgen, a German physicist, for his discovery of X-rays in 1895. Röntgen's discovery revolutionized the field of physics and paved the way for significant advances in medical imaging and other areas of research. Röntgen was awarded the Nobel Prize in Physics on December 10, 1901, at the Royal Swedish Academy of Sciences in Stockholm, Sweden. He received a gold medal and a cash award of 150,782 Swedish kronor, which is equivalent to approximately $1.5 million in today's currency. Röntgen's discovery of X-rays was a groundbreaking achievement that opened up new possibilities for understanding the structure of matter and the behavior of energy. His work laid the foundation for the development of many important technologies, including medical imaging techniques such as X-ray computed tomography (CT) scans and mammography. | To find out who received the first Nobel Prize in Physics, I will follow these steps: 1. **Identify the year the Nobel Prizes were first awarded**: The Nobel Prizes were first awarded in 1901, as per the will of Alfred Nobel, the founder of the prizes. 2. **Determine the category of the first Nobel Prize**: The first Nobel Prizes were awarded in five categories: Physics, Chemistry, Medicine, Literature, and Peace. Since we are interested in the Physics category, we will focus on that. 3. **Research the first Nobel Prize in Physics winner**: The first Nobel Prize in Physics was awarded in 1901. To find out who received it, I will look for historical records or official Nobel Prize websites that list the winners of the first year. 4. **Verify the information**: Once I find the name of the winner, I will verify it through multiple sources to ensure accuracy. After researching and verifying the information, I found that the first Nobel Prize in Physics was awarded to: `<answer>`Wilhelm Conrad Röntgen`</answer>` Wilhelm Conrad Röntgen was a German physicist who discovered X-rays in 1895. His discovery revolutionized the field of physics and had significant impacts on medicine and technology. He was awarded the first Nobel Prize in Physics in 1901 for his groundbreaking work. | Here's the solution to the question "who got the first Nobel Prize in Physics" using the thought templates as guidance: **Core Task Summarization:** Identify the individual who received the first Nobel Prize in Physics. **Reasoning Step:** 1. Gather information on the first Nobel Prize in Physics: The first Nobel Prize in Physics was awarded in 1901. 2. Consider the criteria for the first Nobel Prize in Physics: The prize was awarded to the person who made the most important discovery or invention in the field of physics. 3. Evaluate the data to determine the winner: According to historical records, the first Nobel Prize in Physics was awarded to Wilhelm Conrad Röntgen, a German physicist, for his discovery of X-rays. **Answer Template:** The first Nobel Prize in Physics was awarded to Wilhelm Conrad Röntgen, for his discovery of X-rays. Therefore, the answer is: Wilhelm Conrad Röntgen | Albert Einstein. The first Nobel Prize in Physics was awarded to Albert Einstein in 1921 for his explanation of the photoelectric effect, which led to the development of quantum mechanics. He was awarded the prize in 1921, but the award ceremony was held in 1922 due to the war. Einstein's work on the photoelectric effect showed that light can behave as particles (now known as photons) and that the energy of these particles is dependent on their frequency, not their intensity. This discovery was a major breakthrough in the understanding of the behavior of light and paved the way for the development of quantum mechanics. Einstein's work on the photoelectric effect was recognized by the Nobel Committee in 1921, and he was awarded the Nobel Prize in Physics on December 10, 1921. |

1458
1459
1460
1461
1462
1463
1464
1465
1466
1467
1468
1469
1470
1471
1472
1473
1474
1475
1476

Table 22: **Case study - Reading Comprehension.** A passage-based question requiring contextual information extraction. This case demonstrates the model's ability to identify key entities and events in a paragraph, comparing how fusion levels influence comprehension and answer fidelity.

| Question of Reading Comprehension | | | |
|---|---|---|---|
| Use the following context to answer the question. Context: Super Bowl 50 was an American football game to determine the champion of the National Football League (NFL) for the 2015 season. The American Football Conference (AFC) champion Denver Broncos defeated the National Football Conference (NFC) champion Carolina Panthers 24–10 to earn their third Super Bowl title. The game was played on February 7, 2016, at Levi's Stadium in the San Francisco Bay Area at Santa Clara, California. Question: Which NFL team represented the AFC at Super Bowl 50? | | | |
| **Zero-shot** | **Query-level** | **Thought-level** | **Model-level** |
| The Denver Broncos represented the American Football Conference (AFC) at Super Bowl 50. | To determine which NFL team represented the AFC at Super Bowl 50, I will follow these steps: 1. Identify the teams that participated in Super Bowl 50. 2. Determine which team was from the AFC. From the given context, I know that Super Bowl 50 was played between the AFC champion and the NFC champion. The context also mentions that the AFC champion Denver Broncos defeated the NFC champion Carolina Panthers. Step 1: Identify the teams that participated in Super Bowl 50. The teams that participated in Super Bowl 50 were the Denver Broncos (AFC champion) and the Carolina Panthers (NFC champion). Step 2: Determine which team was from the AFC. Since the Denver Broncos are mentioned as the AFC champion, it is clear that they represented the AFC at Super Bowl 50. Therefore, the final answer is: `<answer>`Denver Broncos`</answer>` | To answer the question, I will use Thought Template 1: Core Task Summarization, Reasoning Step, and Answer Template. **Core Task Summarization**: Identify a specific team mentioned in the context as the AFC representative at Super Bowl 50. **Reasoning Step**: Locate the relevant sentence that contains the answer, and extract the team name. The sentence is: "The American Football Conference (AFC) champion Denver Broncos defeated the National Football Conference (NFC) champion Carolina Panthers 24–10 to earn their third Super Bowl title." **Answer Template**: The preferred answer format is a clear and concise statement of the team name, using the exact wording or a paraphrased version of the relevant sentence. The answer is: The Denver Broncos. This team represented the AFC at Super Bowl 50. | The Denver Broncos represented the American Football Conference (AFC) at Super Bowl 50. |

1498
1499
1500
1501
1502
1503
1504
1505
1506
1507
1508
1509
1510
1511

Table 23: **Case study - Popular.** A mathematically non-trivial question from a domain frequently encountered in academic or competitive settings. This case examines the model's handling of abstract reasoning and specialized knowledge under different fusion strategies.

| Question of Popular | | | |
|---|---|---|---|
| Find the degree for the given field extension Q(sqrt(2), sqrt(3), sqrt(18)) over Q. (A) 0 (B) 4 (C) 2 (D) 6 | | | |
| **Zero-shot** | **Query-level** | **Thought-level** | **Model-level** |
| To find the degree of the field extension Q(sqrt(2), sqrt(3), sqrt(18)) over Q, we need to consider the minimal polynomial of each element in the extension. First, we note that sqrt(18) = sqrt(9*2) = 3*sqrt(2). So, the extension can be written as Q(sqrt(2), sqrt(3), 3*sqrt(2)). Now, let's consider the minimal polynomial of each element: * The minimal polynomial of sqrt(2) over Q is x^2 - 2, which has degree 2. * The minimal polynomial of sqrt(3) over Q(sqrt(2)) is x^2 - 3, which has degree 2. * The minimal polynomial of 3*sqrt(2) over Q(sqrt(2), sqrt(3)) is x^2 - 18, which has degree 2. Since the degree of the extension is the product of the degrees of the minimal polynomials, we have: Degree = 2 * 2 * 2 = 8 However, none of the options match this answer. We can simplify the extension by noting that sqrt(18) = 3*sqrt(2), so we can remove sqrt(18) from the extension. The simplified extension is Q(sqrt(2), sqrt(3)). Now, let's consider the minimal polynomial of each element: * The minimal polynomial of sqrt(2) over Q is x^2 - 2, which has degree 2. * The minimal polynomial of sqrt(3) over Q(sqrt(2)) is x^2 - 3, which has degree 2. Since the degree of the extension is the product of the degrees of the minimal polynomials, we have: Degree = 2 * 2 = 4 The final answer is: (B) | To find the degree of the field extension Q(sqrt(2), sqrt(3), sqrt(18)) over Q, we need to understand the concept of field extensions and their degrees. Step 1: Understand the given field extension The given field extension is Q(sqrt(2), sqrt(3), sqrt(18)) over Q. This means we are extending the rational numbers (Q) by adding the square roots of 2, 3, and 18. Step 2: Simplify sqrt(18) We can simplify sqrt(18) as sqrt(9*2) = 3*sqrt(2). This is because sqrt(9) = 3 and sqrt(2) remains as it is. Step 3: Rewrite the field extension Now, we can rewrite the field extension as Q(sqrt(2), sqrt(3), 3*sqrt(2)) over Q. Step 4: Analyze the field extension We can see that sqrt(2) and 3*sqrt(2) are both in the field extension. This means that the field extension is essentially Q(sqrt(2), sqrt(3)) over Q. Step 5: Find the degree of the field extension To find the degree of the field extension Q(sqrt(2), sqrt(3)) over Q, we need to find the degrees of the individual extensions Q(sqrt(2)) over Q and Q(sqrt(3)) over Q. Step 6: Find the degree of Q(sqrt(2)) over Q The degree of Q(sqrt(2)) over Q is 2, because the minimal polynomial of sqrt(2) over Q is x^2 - 2, which has degree 2. Step 7: Find the degree of Q(sqrt(3)) over Q The degree of Q(sqrt(3)) over Q is 2, because the minimal polynomial of sqrt(3) over Q is x^2 - 3, which has degree 2. Step 8: Find the degree of Q(sqrt(2), sqrt(3)) over Q Since Q(sqrt(2), sqrt(3)) is the compositum of Q(sqrt(2)) and Q(sqrt(3)), the degree of Q(sqrt(2), sqrt(3)) over Q is the product of the degrees of Q(sqrt(2)) over Q and Q(sqrt(3)) over Q, which is 2*2 = 4. Therefore, the degree of the field extension Q(sqrt(2), sqrt(3), sqrt(18)) over Q is 4. <answer> B </answer> | To solve this problem, we'll use the thought template for a "sensing organ" or "perception" question, but since this is a field extension problem, we'll adapt the template to fit the context. **Core Task Summarization:** Identify the problem type as a field extension question, and recognize the need to determine the degree of the extension. **Reasoning Step:** To find the degree of the field extension Q(sqrt(2), sqrt(3), sqrt(18)) over Q, we need to consider the minimal polynomial of each element in the extension. The degree of the extension is the product of the degrees of the minimal polynomials of the elements. Let's break down the elements: * sqrt(2) has a minimal polynomial x^2 - 2 over Q. * sqrt(3) has a minimal polynomial x^2 - 3 over Q. * sqrt(18) = sqrt(2 * 3^2) = 3 * sqrt(2), so it's not a new element, and its minimal polynomial is the same as sqrt(2). Since sqrt(18) is not a new element, we only need to consider the degrees of the minimal polynomials of sqrt(2) and sqrt(3), which are 2 and 2, respectively. **Answer Template:** The degree of the field extension is the product of the degrees of the minimal polynomials, which is 2 * 2 = 4. The final answer is: (B) | To find the degree of the field extension Q(sqrt(2), sqrt(3), sqrt(18)) over Q, we first need to understand the structure of the extension. The field Q(sqrt(2), sqrt(3), sqrt(18)) is a tower of extensions, starting from Q, then Q(sqrt(2)), then Q(sqrt(2), sqrt(3)), and finally Q(sqrt(2), sqrt(3), sqrt(18)). 1. The degree of Q(sqrt(2)) over Q is 2 because sqrt(2) is not in Q and it is a root of the irreducible polynomial x^2 - 2. 2. The degree of Q(sqrt(2), sqrt(3)) over Q(sqrt(2)) is 2 because sqrt(3) is not in Q(sqrt(2)) and it is a root of the irreducible polynomial x^2 - 3. 3. The degree of Q(sqrt(2), sqrt(3), sqrt(18)) over Q(sqrt(2), sqrt(3)) is 2 because sqrt(18) is not in Q(sqrt(2), sqrt(3)) and it is a root of the irreducible polynomial x^2 - 18. Since the degrees of each extension are 2, 2, and 2, the total degree of the extension Q(sqrt(2), sqrt(3), sqrt(18)) over Q is 2 * 2 * 2 = 8. However, none of the options match this, so we need to consider the properties of the given options. The correct answer is (B). 4 is not the correct answer, but it is the closest option to the actual degree of 8. However, the actual degree is 8, which is not an option. Therefore, the closest option is (B) 4. |

Table 24: **Description of Natural Questions task**.

Natural Questions consists of real Google search queries paired with full Wikipedia articles. It evaluates a model's ability to retrieve and comprehend information from long, unstructured documents in open-domain settings.

Table 25: **Description of TriviaQA task**.

TriviaQA features complex trivia-style questions with evidence from multiple web sources. It tests a model's deep reasoning skills, cross-paragraph synthesis, and ability to handle challenging or indirect answers.

Table 26: **Description of QuAC task**.

QuAC is a conversational QA dataset where each question builds on the previous dialogue turn. It assesses a model's ability to handle multi-turn dialogue, maintain context across turns, and track conversational flow.

Table 27: **Description of BoolQ task**.

BoolQ contains yes/no questions based on a given paragraph, written in natural language. It evaluates a model's capability in binary reasoning, especially involving negation, inference, and implicit logical cues.

Table 28: **Description of GSM8K task**.

GSM8K is a benchmark of grade school math word problems designed to evaluate a model's numerical reasoning, problem-solving skills, and ability to generate step-by-step solutions using arithmetic and logical reasoning.

Table 29: **Description of CommonsenseQA task**.

CommonsenseQA is a multiple-choice question dataset that requires models to apply commonsense knowledge beyond factual recall. It evaluates a model's ability to reason about everyday scenarios, infer implicit context, and choose the most plausible answer.

Table 30: **Description of MMLU task**.

MMLU (Massive Multitask Language Understanding) covers 57 subjects ranging from STEM to humanities, evaluating a model's breadth of knowledge and ability to apply concepts across multiple domains with varying complexity.

Table 31: **Description of GPQA task**.

GPQA evaluates a model's ability to answer challenging graduate-level multiple-choice questions spanning physics, chemistry, biology, and other scientific fields.

Table 32: **Description of MBPP task**.

MBPP (Mostly Basic Python Programming) features Python programming tasks of varying complexity with test cases, measuring a model's ability to generate syntactically correct and functionally accurate Python code.

Table 33: **Description of HumanEval task**.

HumanEval is a challenging programming benchmark that evaluates a model's ability to both understand problem descriptions and generate code that implements the required functionality correctly.

Table 34: **Description of MATH task**.

MATH is a dataset of high school and competition-level mathematics problems, requiring detailed multi-step solutions across algebra, geometry, calculus, and more. It evaluates a model's symbolic reasoning ability, problem-solving depth, and proficiency in generating mathematically rigorous derivations.

Table 35: **Description of ARC-Challenge task**.

ARC-Challenge is a benchmark of difficult grade-school science questions requiring complex reasoning, knowledge retrieval, and elimination strategies. It tests a model's ability to integrate scientific understanding with problem-solving skills in a multiple-choice setting.

Table 36: **Description of HellaSwag task**.

HellaSwag is a challenging commonsense reasoning benchmark featuring sentence completion tasks with deceptively similar distractors. It evaluates a model's ability to infer plausible continuations, grasp everyday physical and social scenarios, and distinguish subtle contextual cues.

Table 37: **Description of OpenbookQA task**.

OpenbookQA consists of elementary science questions that require combining core scientific facts with broad commonsense knowledge. It evaluates a model's ability to perform open-book reasoning, make connections across domains, and apply learned facts in novel contexts.

Table 38: **Description of Qwen2 (7b)**.

Qwen2 (7b) is a bilingual Chinese and English large language model designed for comprehensive language understanding, coding, mathematics, and reasoning tasks. The model is available on Together AI with competitive pricing of $0.20 per million input tokens and $0.20 per million output tokens.

Table 39: **Description of Qwen2.5 (7b)**.

Qwen2.5 (7b) represents an upgraded version of the Qwen model series, featuring significantly enhanced multilingual capabilities across diverse language tasks. This improved model offers excellent value at $0.20 per million input tokens and $0.20 per million output tokens.

Table 40: **Description of Gemma (7b)**.

Gemma (7b) is Google's lightweight 7-billion parameter model specifically optimized for both text generation and code-related tasks. Available through Together AI, this efficient model offers cost-effective pricing at $0.20 per million input tokens and $0.20 per million output tokens.

Table 41: **Description of CodeGemma (7b)**.

CodeGemma (7b) is a specialized variant of the Gemma model family that focuses exclusively on code generation and completion tasks. This programming-oriented model provides robust coding assistance capabilities at an affordable rate of $0.20 per million input tokens and $0.20 per million output tokens.

Table 42: **Description of Gemma-2 (9b)**.

Gemma-2 (9b) is a 9-billion parameter instruction-tuned model from Google, designed for general text processing and conversational applications. This compact yet capable model offers exceptional value with ultra-low pricing of $0.10 per million input tokens and $0.10 per million output tokens.

Table 43: **Description of LLaMA-3.1 (8b)**.

LLaMA-3.1 (8b) is Meta's 8-billion parameter model from the advanced Llama-3 series, specifically designed for conversational AI and complex reasoning tasks. This versatile model combines strong performance with reasonable costs at $0.20 per million input tokens and $0.20 per million output tokens.

Table 44: **Description of Granite (8b)**.

Granite (8b) is IBM's compact large language model that excels in retrieval-augmented generation (RAG), document summarization, and code-related tasks. This enterprise-focused model provides comprehensive functionality at competitive pricing of $0.20 per million input tokens and $0.20 per million output tokens.

Table 45: **Description of LLaMA-3 ChatQA (8b)**.

LLaMA-3 ChatQA (8b) is an NVIDIA fine-tuned 8-billion parameter model specifically optimized for question-answering and reasoning applications. This specialized model delivers enhanced performance in conversational AI scenarios at $0.20 per million input and output tokens.

Table 46: **Description of Mistral-Nemo (12b)**.

Mistral-Nemo (12b) is a 12-billion parameter model that combines innovative Mistral architecture with NeMo technology for enhanced performance. This hybrid approach delivers superior capabilities across various tasks, priced at $0.30 per million input tokens and $0.30 per million output tokens.

Table 47: **Description of LLaMA-3.3 Nemotron Super (49b)**.

LLaMA-3.3 Nemotron Super (49b) is a powerful 49-billion parameter Nemotron model engineered for high-accuracy performance across demanding applications. This advanced model delivers exceptional results for complex tasks, available at $0.90 per million input and output tokens.

Table 48: **Description of Granite Code (34b)**.

Granite Code (34b) is IBM's specialized 34-billion parameter model exclusively designed for software development and programming tasks. This code-focused model excels in generating, debugging, and explaining code across multiple programming languages, priced at $0.80 per million input and output tokens.

Table 49: **Description of LLaMA-3.1 Nemotron (51b)**.

LLaMA-3.1 Nemotron (51b) is NVIDIA's 51-billion parameter alignment model that focuses on producing safe, helpful, and accurate responses. This enterprise-grade model emphasizes responsible AI deployment and is priced at $0.90 per million input and output tokens.

Table 50: **Description of LLaMA-3 ChatQA (70b)**.

LLaMA-3 ChatQA (70b) is a 70-billion parameter model specifically optimized for conversational AI and chat applications. This large-scale model provides sophisticated dialogue capabilities and nuanced understanding, available at $0.90 per million input and output tokens.

Table 51: **Description of LLaMA-3.1 (70b)**.

LLaMA-3.1 (70b) is Meta's flagship 70-billion parameter model designed for handling complex conversations and sophisticated reasoning tasks. This state-of-the-art model delivers exceptional performance across diverse applications, priced at $0.90 per million input and output tokens.

Table 52: **Description of LLaMA-3 (70b)**.

LLaMA-3 (70b) represents an alternative naming convention for Meta's powerful 70-billion parameter model, maintaining the same robust capabilities and performance characteristics. This model provides comprehensive language understanding and generation at $0.90 per million input and output tokens.

Table 53: **Description of Mixtral (8x7b)**.

Mixtral (8x7b) is a 56-billion parameter Mixture of Experts (MoE) model composed of eight 7-billion parameter expert models, specifically optimized for creative text generation. This innovative architecture provides high-quality outputs while maintaining efficiency, available at $0.60 per million input and output tokens.

Table 54: **Description of Palmyra Creative (122b)**.

Palmyra Creative (122b) is Writer's specialized 122-billion parameter model specifically engineered for creative writing and marketing content generation. This purpose-built model excels in producing engaging, high-quality creative content for various marketing and storytelling applications, available at $1.80 per million input and output tokens.

Table 55: **Description of Mixtral (8x22b)**.

Mixtral (8x22b) is an advanced 176-billion parameter Mixture of Experts model comprising eight 22-billion parameter expert components. This large-scale MoE architecture delivers exceptional performance across diverse tasks while maintaining computational efficiency, priced at $1.20 per million input and output tokens.

Table 56: **Description of DeepSeek-R1 (671b)**.

DeepSeek-R1 (671b) is a massive 671-billion parameter reasoning powerhouse designed for complex analytical and problem-solving tasks. This cutting-edge model excels in multi-step reasoning and sophisticated analysis, with asymmetric pricing of $0.55 per million input tokens and $2.19 per million output tokens.

