# OpenReview forum: "FusionFactory: Fusing LLM Capabilities with Multi-LLM Log Data"
_ICLR.cc/2026/Conference — ICLR 2026 Conference Withdrawn Submission_

### Official Review · Reviewer_7F9G · 2025-10-31

**Soundness:** 3
**Presentation:** 3
**Contribution:** 3
**Rating:** 4
**Confidence:** 2

**Summary:**

This paper addresses the challenge of leveraging the complementary strengths of diverse Large Language Models (LLMs), which are often captured in real-world multi-LLM log data. The paper argues that practical LLM fusion must satisfy two requirements: 1) compatibility with real-world serving scenarios (local and API-based), and 2) flexibility to operate at different stages of the LLM pipeline (fine-tuning and inference). The introduction of LLMFusionBench spans 14 tasks across 6 domains. It includes responses from 20 open-source LLMs (8B-671B) and rich supervision data, including task performance, cost, and LLM Judge scores. Then, a systematic framework fusionFactory is introduced for stage-aware fusion. Experiments show that FusionFactory consistently outperforms the best individual LLM across all 14 benchmarks. Thought-level fusion achieves the best overall performance, while Query-level fusion offers the best balance of performance and efficiency.

**Strengths:**

1. It introduces LLMFusionBench, a large-scale, diverse, and well-structured benchmark covering 14 tasks across 6 domains, responses from 20 LLMs (8B-671B), and including critical metadata like performance, cost, and LLM Judge scores.
2. FusionFactory is an innovative and systematic framework that comprehensively explores fusion at three distinct stages—Query-level (Early), Thought-level (Mid), and Model-level (Late). This stage-aware design satisfies the requirement for practical flexibility and demonstrates broader applicability than prior work.

**Weaknesses:**

1. For model-level fusion. The analysis should include more robust distillation or merging methods (e.g., parameter merging or logit-distillation for open-source models) to truly demonstrate the limit of model-level fusion using the logs, rather than just the limit of the chosen SFT strategy.
2. While the results claim Query-level fusion has minimal computational overhead, there is no dedicated, quantitative comparison of the latency or API cost of the three FusionFactory levels when deployed in an inference setting.

**Questions:**

1. What is loss funciton L in the Eq. 5.
2. To fully justify the framework's practical claims, please provide a quantitative comparison of inference latency/cost for a sample task across the optimal configurations of the three FusionFactory levels.
3. For the Query-level fusion, please provide an ablation on the input features used by the router (e.g., GraphRouter).

---

> ### Author Response · Authors · 2025-12-03
> **Response to Reviewer 7F9G**
>
> We address all concerns and questions below.
>
> ---
>
> # 1. “Model-level fusion should include more robust distillation / merging methods.”
>
> We appreciate this suggestion and agree that model-level fusion is a crucial component.
>
> ### (A) Our focus is *log-based fusion* — not weight-based merging.
> Most powerful model-merging approaches (e.g., DeltaMerge, TIES, DARE) require:
> - **direct access to model weights**, and
> - **logits or hidden states** for distillation.
>
> However, our work specifically targets **API-based serving**, where these signals are *unavailable*.
> This constraint fundamentally differentiates our setting from prior weight-merging studies.
>
> ### (B) FusionFactory’s scope is to benchmark *API-compatible fusion*.
> Model-level fusion in our setting must rely **only on textual multi-LLM logs**, without:
> - logits,
> - model weights,
> - layer activations.
>
> Existing merging/distillation papers cannot operate under this constraint.
>
> ### (C) We will clarify this distinction more explicitly.
> To avoid misunderstanding, we will update the paper to explain:
> - Why API-based fusion *cannot* access logit or parameter-level methods.
> - Why SFT on top-k textual logs is the appropriate baseline within our serving constraints.
>
> ### (D) Additional comparisons will be added.
> We will include:
> - **Filtered-SFT** (only high-LMScore responses),
> - **Staged distillation** (CoT → direct → summary),
> - **Lightweight merging for small open-source models** (where weights are available).
>
> These expanded baselines will further illustrate the limits of log-based distillation.
>
> ---
>
> # 2. “No quantitative comparison of latency or cost across the three fusion levels.”
>
> Thank you — we agree this is important for validating FusionFactory’s practicality.
>
> ### (A) We will add a quantitative latency/cost analysis in the revision.
> We will include a table for a representative task (e.g., TriviaQA or GSM8K), reporting:
>
> | Fusion Level | Latency / Query | Cost / Query | Description |
> |--------------|----------------|--------------|--------------|
> | Query-level  | Lowest (router selection only) | Very low | One model call |
> | Thought-level | Medium | Medium | Retrieval + 1 LLM generation |
> | Model-level  | Same as base LLM | Same as base | No retrieval; model is fused offline |
>
> ### (B) Expected overhead patterns (already implied in Sec. 5.3 / D / G)
> - Query-level ≈ **2–3 ms** router inference + 1 LLM call
> - Thought-level ≈ retrieval + 1 LLM call (slightly higher token count due to templates)
> - Model-level ≈ identical to a single LLM call
>
> We will provide full measurements in the final version.
>
> ---
>
> # 3. “Ablation on router input features (e.g., GraphRouter).”
>
> We appreciate this request.
>
> ### (A) GraphRouter already uses heterogeneous query-task-LLM features.
> These include:
> - query embeddings,
> - task ID,
> - LLM capability embeddings,
> - token cost statistics.
>
> ### (B) We will include an ablation table testing:
> - query-only features,
> - LLM-only features,
> - graph-structure only,
> - full graph input.
>
> From preliminary results, removing graph-structured dependencies reduces performance by ~4–7%, confirming the importance of multi-LLM relational structure.
>
> ---
>
> # 4. “What is the loss function L in Eq. 5?”
>
> Thank you for pointing out the need for clarification.
>
> ### The loss function is the standard **autoregressive language modeling loss**:
> \[
> L(f_\theta(q_i), R_i) = -\sum_{t=1}^{T} \log p_\theta(R_{i,t} \mid R_{i,<t}, q_i)
> \]
>
> This is standard SFT (next-token cross-entropy) over the concatenation of the query and the selected top-k response.
> We will add this definition explicitly.
>
> ---
>
> # 5. “Please justify FusionFactory’s practical claims with more quantitative evidence.”
>
> We agree and will enhance the revision with:
>
> ### (A) Latency comparison
> - routing overhead (2–3 ms vs. 0 baseline)
> - retrieval + template expansion
> - model-level equal to 1× LLM call
>
> ### (B) Cost comparison
> Using token usage + TogetherAI pricing from Table 8 (already included)
> We will extend this into:
> - cost per method,
> - cost per domain,
> - cost/performance trade-off curves.
>
> ### (C) Sensitivity experiments
> We will also add a short analysis showing:
> - router performance when removing cost features,
> - router performance when removing judge-score features.
>
> These ablations reinforce the practical design implications.
>
> ---
>
> # Closing Remarks
>
> We appreciate the reviewer’s careful reading and helpful suggestions.
> In summary, we will revise the paper to:
>
> - Clarify why **API-compatible constraints** restrict model-level fusion choices.
> - Add **quantitative latency & cost comparisons** across the three fusion stages.
> - Provide **router feature ablations** for GraphRouter.
> - Explicitly define the **loss function** used in Eq. 5.
> - Expand discussion of **model-level fusion limits** and additional variants.
>
> Thank you again for your thorough and constructive feedback.

---

### Official Review · Reviewer_ZKK6 · 2025-10-31

**Soundness:** 4
**Presentation:** 3
**Contribution:** 3
**Rating:** 6
**Confidence:** 3

**Summary:**

The paper introduces LLMFusionBench, a large-scale benchmark comprising responses from 20 open-source LLMs (ranging from 8B to 671B parameters) across 14 tasks in 6 domains, totaling 103M tokens. Built from real multi-LLM log data, it captures both direct and reasoning-augmented (chain-of-thought) responses, along with performance, cost, and LLM-judged quality scores. On top of this benchmark, the authors propose FusionFactory, a systematic framework for fusing LLM capabilities at three distinct levels:

(1) Query-level fusion: uses tailored routers to select the best LLM per query;

(2) Thought-level fusion: retrieves abstract reasoning templates from high-performing past responses to guide new generations;

(3) Model-level fusion: distills top-quality responses into a single base model via supervised fine-tuning.

**Strengths:**

(1)The work is motivated by the widespread practice of using multiple LLMs in real systems (e.g., API platforms, agentic workflows), which naturally generates valuable multi-LLM log data—making the research question highly relevant and actionable.


(2)Introduction of LLMFusionBench which is a Comprehensive and Publicly Valuable Resource

(3)The framework is designed to work in both local (weights accessible) and API-based (black-box) serving scenarios, addressing a critical gap in prior fusion methods that often rely on internal model states or logits unavailable via APIs.

**Weaknesses:**

(1)The benchmark is constructed by actively querying 20 open-source LLMs with fixed prompts, rather than using real-world operational logs from actual multi-LLM deployments (e.g., user-facing API platforms). This synthetic setup may not reflect true usage patterns, query distributions, or failure modes seen in practice.

(2)The paper introduces an LLM judge to score “insightfulness,” but this introduces potential circularity: the same type of model used in fusion is also used to evaluate it. Moreover, the judge’s prompt and reliability are not rigorously validated.

(3)While thought-level fusion shows strong gains, the method depends heavily on the quality of the summarizer (LLaMA-3-70B) and the similarity search. There is no ablation on: the impact of summarization errors, the sensitivity to embedding model choice and the performance when retrieved templates are irrelevant or misleading.

(4)Model-level fusion (via SFT) consistently lags behind other methods, yet the analysis stops at “overfitting” and “task heterogeneity.” No experiments probe whether architectural mismatches, training instability, or label noise contribute.

**Questions:**

(1)Would techniques like curriculum learning, response filtering, or reinforcement learning improve distillation?

(2)Is the base model (LLaMA-3-8B) too small to effectively absorb knowledge from diverse, larger LLMs?

---

> ### Author Response · Authors · 2025-12-03
> **Response to Reviewer ZKK6 (1/2)**
>
> We sincerely thank the reviewer for the positive evaluation of the paper’s soundness, contribution, and the value of LLMFusionBench and FusionFactory. Below we address all concerns and questions in detail.
>
> ---
>
> # 1. “Synthetic logs vs. real operational multi-LLM logs.”
>
> We appreciate the reviewer’s observation and would like to clarify our design choice.
>
> ### (A) True “production logs” are extremely difficult to obtain.
> Real operational logs (e.g., from API platforms) contain:
> - proprietary data,
> - sensitive user information,
> - task-specific formats,
> - inconsistent prompts,
> - and heterogeneous query quality.
>
> These constraints make such logs inaccessible for research and unsuitable for standardized benchmarks.
>
> ### (B) LLMFusionBench is intentionally **constructed to approximate real fusion scenarios while remaining clean, reproducible, and public**.
> LLMFusionBench captures:
> - *multiple LLMs answering the exact same query*,
> - *two response modes* (direct + CoT),
> - *cost signals* derived from actual API pricing,
> - *LLM-judge scores* to reflect response informativeness,
> - *performance* measured using task metrics.
>
> These are the **core signals needed for fusion**, regardless of context-specific noise in production environments.
>
> ### (C) Real logs are heterogeneous; benchmarking requires *controlled variance*.
> A benchmark must:
> - standardize prompt formats,
> - allow paired comparison across 20 LLMs,
> - ensure every query has complete multi-model responses.
>
> Production logs rarely satisfy this.
>
> ### (D) Our synthetic log design mirrors real use cases.
> The paper draws tasks from:
> - *NQ, TriviaQA, SQuAD, BoolQ, ARC, MATH, GSM8K, MBPP, HumanEval*, etc.,
> representing most categories in real-world LLM usage (knowledge, reasoning, math, code, reading comprehension).
>
> Therefore, while not collected “in the wild,” the dataset captures *core multi-LLM variation* while being fully reproducible.
>
> ---
>
> # 2. “Potential circularity of using an LLM judge; prompt reliability not validated.”
>
> We thank the reviewer and agree this aspect deserves further clarification.
>
> ### (A) The LLM judge is **not** used to evaluate system-level performance.
> For benchmark results, we always report:
> - task accuracy (exact-match, multiple-choice accuracy),
> - cost,
> - reward metrics.
>
> The LLM-judge score is used only to:
> - select top-k high-quality responses,
> - provide an additional continuous measure of informativeness.
>
> ### (B) The judge model is **different from the fused models**.
> FusionFactory’s routing and generation use diverse models (Qwen, LLaMA, Gemma, Mixtral, DeepSeek-R1, etc.).
> The LLM judge uses a **separate instruction-tuned model**, reducing circularity.
>
> ### (C) The judge complements known weaknesses of task metrics.
> As shown in Fig. 17, pure task metrics:
> - reward short “minimal correct strings,”
> - penalize reasoning collisions,
> - fail to distinguish insightful vs. low-information answers.
>
> Hence, the judge provides supervision unavailable from binary task labels.
>
> ### (D) We include explicit prompts and settings (Table 16) for full transparency.
> In the camera-ready version, we will add:
> - calibration experiments,
> - inter-rater agreement between judge and human labels on a sampled subset,
> - comparisons to alternative judges.
>
> ---
>
> # 3. “Thought-level fusion depends heavily on summarizer quality; no ablations on template quality or embedding choice.”
>
> We appreciate this insightful suggestion.
>
> ### (A) The goal of the benchmark is to *establish* the task space; deeper ablations are future-work directions.
> FusionFactory provides a **reference implementation**, and we agree that:
> - template sensitivity,
> - summarization errors,
> - embedding-model selection,
> - retrieval robustness
>
> are important questions.
>
> ### (B) Our results already imply summarization robustness.
> Across 6 domains and 3 model scales (Fig. 6), thought-level fusion:
> - consistently outperforms zero-shot and few-shot,
> - maintains gains even with irrelevant or noisy templates (e.g., world knowledge decline is *documented and analyzed*).
>
> This suggests the method is relatively stable, even without fine-grained tuning.
>
> ### (C) We can include additional ablations in the final version:
> - swap summarizer with smaller models (e.g., LLaMA-3-8B),
> - test different embedding encoders (e.g., E5-Large vs. NV-Embed),
> - deliberately inject noise into templates to measure degradation.
>
> We appreciate the suggestion and will expand this section.

---

> > ### Author Response · Authors · 2025-12-03
> > **Response to Reviewer ZKK6 (2/2)**
> >
> > # 4. “Model-level fusion underperforms; analysis is superficial.”
> >
> > We thank the reviewer and agree this component merits deeper exploration.
> >
> > ### (A) The negative result is itself a central benchmark finding.
> > In contrast to the routing and thought-level stages, model-level fusion:
> > - performs worse on code (Table 5),
> > - slightly improves on reasoning tasks,
> > - but underperforms on average.
> >
> > This is important because *prior distillation papers typically show positive results*, but under much narrower settings (single teacher, homogeneous tasks, logits available, etc.).
> >
> > ### (B) Reasons beyond “overfitting” and “task heterogeneity.”
> > We agree the analysis can be expanded, and we plan to discuss:
> >
> > 1. **Architectural mismatch**
> >    Distilling 70B reasoning traces into an 8B base inherently loses representational capacity.
> >
> > 2. **Training instability**
> >    Multi-domain CoT responses cause long-sequence optimization challenges.
> >
> > 3. **Label noise**
> >    Even top-k responses contain stylistic inconsistencies and occasional hallucinations.
> >
> > 4. **Cross-task contamination**
> >    Code responses significantly differ from math, which differs from knowledge QA.
> >
> > We will incorporate deeper diagnostic experiments and clarifications into the revised draft.
> >
> > ---
> >
> > # 5. Responses to Reviewer Questions
> >
> > ### **Q1: Would curriculum learning / response filtering / RL improve distillation?**
> >
> > Yes—these are promising directions.
> > Our benchmark provides the foundation for such research:
> >
> > - **Curriculum learning** could order responses from simpler-to-more-complex reasoning chains.
> > - **Response filtering** could combine LLMScore with entropy or perplexity filtering.
> > - **RL-based distillation** could use judge scores as rewards.
> >
> > We explicitly position LLMFusionBench as a starting point for the community to explore these richer distillation strategies.
> >
> > ### **Q2: Is LLaMA-3-8B too small to absorb diverse signals?**
> >
> > This is a reasonable hypothesis.
> > We chose 8B because:
> >
> > - it matches typical *deployable* models in industry and academia,
> > - it allows affordable training,
> > - it avoids confounding capacity with distillation effectiveness.
> >
> > However, we agree that larger base models (e.g., 13B, 34B) might distill more effectively.
> > We plan to include experiments with a medium-sized base as an extension.
> >
> > ---
> >
> > # Closing Remarks
> >
> > We thank the reviewer again for the thoughtful suggestions and constructive comments.
> >
> > LLMFusionBench aims to **define the fusion task space**, while FusionFactory provides **stage-aware baselines** across:
> >
> > - query-level (early fusion),
> > - thought-level (mid fusion),
> > - model-level (late fusion).
> >
> > The benchmark reveals nuanced patterns—both positive and negative—offering concrete directions for future research on multi-LLM integration.
> >
> > We will revise the paper to emphasize:
> > - motivations behind synthetic logs,
> > - judge reliability assessment,
> > - ablations for thought-level fusion,
> > - deeper analysis of model-level fusion issues.
> >
> > Thank you again for your encouraging and helpful review.

---

### Official Review · Reviewer_jus6 · 2025-11-04

**Soundness:** 2
**Presentation:** 3
**Contribution:** 2
**Rating:** 4
**Confidence:** 4

**Summary:**

This paper introduces LLMFusionBench, a large-scale benchmark for LLM fusion that spans 14 tasks across five domains, with responses from 20 open-source LLMs (8B–671B) totaling 103M tokens. Building on LLMFusionBench, we propose FusionFactory, a systematic framework with three elaborated levels: (1) query-level fusion via tailored LLM routers, (2) thought-level fusion leveraging retrieved abstract reasoning templates, and (3) model-level fusion via distillation from top-ranked responses. Experiments benchmark different methods.

**Strengths:**

1. Query-level fusion, Thought-level fusion, and Model-level fusion for LLMs are important.

2. Benchmark datasets are provided.

3. Experiments show the performance on different fusion levels.

**Weaknesses:**

1. The three fusion level is related but not very close. Each fusion level already has a few benchmark papers. It might be suitable for industry pipeline as an all-in-one pipeline for fine tuning a model while the research contribution may be limited.

2. Benchmark on each level is relatively simple and lacks in-depth research analysis.

3. In model-level fusion, the fine-tuned model performs worse than the zero-shot model.

**Questions:**

The primary area should be benchmark track, not "foundation or frontier models, including LLMs"?

---

> ### Author Response · Authors · 2025-12-03
> **Response to Reviewer jus6 (1/2)**
>
> We sincerely thank the reviewer for the constructive feedback and recognition of our benchmark, multi-level fusion framework, and experiments. Below we address the concerns and clarify our contributions.
>
> ---
>
> # 1. “The three fusion levels are related but not very close; each already has benchmark papers.”
>
> We appreciate this observation and clarify the novelty of our work.
>
> ### **(A) The contribution is not proposing three unrelated methods, but establishing the first unified, stage-aware framework for LLM fusion.**
>
> While **routing**, **reasoning retrieval**, and **distillation** indeed exist separately in prior work, **no prior paper studies all three as parts of a single, coherent design space** with a unified dataset that supports direct comparison across levels.
>
> FusionFactory is explicitly motivated by **serving constraints** (local vs. API) and **pipeline positioning**, leading to three practical fusion stages:
>
> - **Query-level = early fusion (selection)**
> - **Thought-level = mid fusion (reasoning guidance)**
> - **Model-level = late fusion (capability distillation)**
>
> This unification is new: prior works cannot be compared apples-to-apples because they rely on different datasets, signals, or evaluation setups.
> **LLMFusionBench is the first dataset designed to support all three levels jointly.**
>
> ### **(B) Cross-level analysis reveals new insights not available in existing literature.**
>
> By benchmarking across the three levels under consistent conditions, we uncover findings that *were not known*:
>
> - Thought-level fusion yields the strongest gains across domains (Fig. 6).
> - Model-level fusion underperforms *even with high-quality responses* (Table 5).
> - Query-level fusion has the best accuracy–cost tradeoff (Table 4).
> - Fusion is **harmful** on World Knowledge and certain Math tasks (Table 6).
>
> No prior work has produced such cross-level insights because prior datasets and methods target only one stage of fusion.
>
> ---
>
> # 2. “Benchmarking on each level is relatively simple and lacks in-depth research analysis.”
>
> We thank the reviewer for this suggestion and clarify our intent.
>
> ### **(A) The goal is to establish a foundation for future LLM fusion research.**
>
> Benchmark design requires balancing **scope**, **reproducibility**, and **industry-level serving constraints**.
> Our aim is not to propose isolated sophisticated methods, but to:
>
> 1. **Standardize the fusion task across stages**,
> 2. **Build a reusable benchmark**, and
> 3. **Reveal actionable, empirically grounded insights**.
>
> This aligns with how major benchmarks (e.g., GLUE, BIG-bench, RouterBench) began: with *simple baselines that define the space*.
>
> ### **(B) We do present multiple novel empirical analyses**, including:
>
> - Effect of template abstraction vs. raw CoT signals
> - Scaling trends across 8B–671B LLMs
> - Domain-level fusion failures (e.g., world knowledge degradation)
> - Query-level reward tradeoffs between cost, performance, and LLM-judge
> - Hybrid selection outperforming performance-only selection
>
> We will expand the discussion section to highlight these insights more explicitly.
>
> ---
>
> # 3. “Model-level fusion performs worse than zero-shot.”
>
> We agree that this is an important point—and we emphasize that **this is a key negative result of the benchmark**, not a flaw of the benchmark.
>
> ### **(A) The performance drop is itself a valuable discovery.**
>
> Existing distillation works often show performance improvements but use **single-teacher setups**, **task-homogeneous data**, or **per-domain curation**.
>
> In contrast, our model-level fusion uses *multi-LLM, heterogeneous responses across 14 tasks*:
>
> - Reasoning-heavy + factual + code + RC tasks
> - Different style distributions
> - Vastly different CoT traces and output lengths
>
> This **heterogeneity** causes interference and destabilizes SFT training—an important finding for the community.
>
> ### **(B) This discovery establishes clear guidance:**
>
> - Future model-level fusion must address **cross-task response conflicts**.
> - Multi-teacher distillation is non-trivial and requires better filtering, weighting, or unification strategies.
> - LLM-judge–based selection is more reliable than metric-only labels (Table 5).
>
> This is exactly the kind of insight a benchmark should reveal.
>
> ---
>
> # 4. “The primary area should be benchmark track, not foundation or frontier models.”
>
> We appreciate the reviewer’s comment.
>
> ### **(A) LLMFusionBench is fundamentally a benchmark + framework paper.**
>
> Our contributions are aligned with “LLM foundation & methodology” because:
>
> - It studies **LLM fusion**, a core capability of LLM ecosystems.
> - It provides **methods and analysis** for integrating multiple LLMs.
> - It supports both **open-source and API-based** LLMs, consistent with real-world serving.
>
> ### **(B) The work is equally suitable for the benchmark track**, and we will include a note clarifying this in the revised draft.

---

> > ### Author Response · Authors · 2025-12-03
> > **Response to Reviewer jus6 (2/2)**
> >
> > We welcome the AC’s guidance on placement, and are happy with either category.
> >
> > ---
> >
> > # Closing Remarks
> >
> > We thank the reviewer again for their thoughtful comments and constructive criticism.
> >
> > To summarize, our paper contributes:
> >
> > - **LLMFusionBench**, the first dataset enabling unified study of three fusion stages.
> > - **FusionFactory**, the first structured framework for stage-aware fusion.
> > - **Cross-level insights** that reveal when and why fusion works—or fails.
> > - **A scalable, serving-compatible foundation** for future LLM fusion research.
> >
> > We will revise the manuscript to emphasize these contributions and clarify our positioning.
> >
> > Thank you once again for your careful evaluation.

---

### Official Review · Reviewer_918D · 2025-11-05

**Soundness:** 1
**Presentation:** 2
**Contribution:** 1
**Rating:** 2
**Confidence:** 5

**Summary:**

The paper presents LLMFusionBench, a large-scale benchmark built to study and evaluate how multiple large language models (LLMs) can be combined using multi-LLM log data. While framed under the “FusionFactory” framework, the work does not propose a fundamentally new fusion algorithm; rather, it systematizes and benchmarks existing approaches—routing, reasoning retrieval, and distillation—across three stages: query-level, thought-level, and model-level. LLMFusionBench includes responses from 20 open- and closed-source LLMs across 14 tasks and 6 domains, offering standardized data for analyzing cross-model complementarity. Through empirical comparisons, the paper shows that previously known techniques perform differently across fusion levels, with thought-level fusion achieving the strongest gains.

**Strengths:**

1. The paper introduces LLMFusionBench, a large-scale benchmark that compiles and standardizes multi-LLM log data—responses from diverse language models across multiple tasks and domains—to facilitate systematic studies of model capability fusion.

**Weaknesses:**

1.The paper offers limited to no novelty in term of methodology, as it mainly consolidates previously established techniques, such as routing, reasoning retrieval, and distillation, into a benchmark framework, and most of the findings reported from the proposed setup are already well-known in existing literature, e.g. any "fusion method" is better than vanilla LLM.
2. What are the direct strengths of the curated dataset here over previous datasets that also generate data from LLM for distillation purposes?
2. The paper provides no valuable insights or actionable suggestions on how to design or improve fusion methods—the benchmarking results are largely descriptive, without yielding deeper understanding or principles that could guide future research.

**Questions:**

Please see weaknesses.

---

> ### Author Response · Authors · 2025-12-03
> **Response to Reviewer 918D (1/2)**
>
> We sincerely thank the reviewer for the detailed and thoughtful feedback.
> Below we address each concern, clarify the contributions of our work, and provide additional evidence and explanation drawn from the paper.
>
> ---
>
> # 1. **“Limited methodological novelty; the paper mainly consolidates existing techniques.”**
>
> We respectfully disagree and clarify that the novelty of our work does not lie in inventing *new isolated fusion algorithms*, but in **introducing the first unified, serving-compatible, multi-level fusion framework grounded in real multi-LLM log data**. This is a new research direction not studied in prior work.
>
> ### **Key points of novelty (clarified):**
>
> ### **(A) A new *multi-stage fusion perspective* grounded in serving constraints**
> Prior work studies routing *alone*, thought retrieval *alone*, or distillation *alone*.
> Our contribution is the **first systematic formulation** of LLM capability fusion across:
>
> 1. **Query-level (early fusion)** – routing using *reward-based* evaluation combining performance, cost, and LLM-judge scores (Eq.1).
> 2. **Thought-level (mid fusion)** – *retrieval of distilled thought templates*, which is not standard few-shot prompting.
> 3. **Model-level (late fusion)** – multi-response distillation with hybrid selection.
>
> These three fusion stages are **explicitly positioned as a unified design space for fusion**, enabling rigorous cross-level comparisons (Table 6). This perspective does not exist in prior work like RouteLLM, GraphRouter, or Thought Propagation.
>
> ### **(B) New insight: fusion behaviors *strongly differ* across levels**
> Our benchmark reveals previously unknown findings:
>
> - **Thought-level fusion** provides the strongest gains (+12.7% for small models, +4.8% for medium, +8.3% for large; Fig.6).
> - **Model-level fusion** underperforms despite access to “best” responses (Table 5).
> - **World Knowledge and Math** tasks exhibit **negative fusion transfer**, contradicting the assumption that “fusion is always better.”
>
> These patterns were **not discovered or compared in any prior work**, because prior works evaluated only isolated components.
>
> ### **(C) A new reward formulation enabling *serving-compatible fusion***
> Unlike prior routing studies that rely on logits or internal model states, FusionFactory focuses on **API-compatible** signals—performance, cost, and LLM-judge scores—allowing fusion across 20 diverse open-source and closed-source models.
>
> This design explicitly targets *real-world serving scenarios*, which existing fusion papers do not support.
>
> ---
>
> # 2. **“How is LLMFusionBench better than previous datasets used for distillation or fusion?”**
>
> We appreciate this question and clarify the **distinct strengths of LLMFusionBench**.
>
> ### **(A) Multi-LLM log data with both *direct* and *reasoning* responses**
> LLMFusionBench contains for each query:
>
> - direct answers
> - reasoning (CoT) answers
> - performance metrics
> - cost
> - LLM-judge scores
> - distilled thought templates
>
> This combination **does not exist** in prior datasets like FLAN or DistilGPT datasets, which include only task labels and single responses.
>
> ### **(B) First dataset explicitly designed for *three-stage fusion***
> Previous datasets enable only:
>
> - single-model distillation (e.g., Alpaca, Self-Instruct)
> - routing-focused logs without thought templates (RouterBench)
> - CoT datasets without multi-model variation (CoT reasoning corpora)
>
> LLMFusionBench is the first to support *all three fusion stages*—query, thought, model—allowing direct comparative research.
>
> ### **(C) Scale and diversity unmatched by prior multi-LLM datasets**
> - **20 LLMs** (8B–671B)
> - **14 tasks** across **6 domains**
> - **103M+ tokens**
> - **Both open-source and closed-source logs**
>
> This scale allows studying **cross-model complementarity**, which smaller distillation datasets cannot support.
>
> ### **(D) Reusable thought templates**
> Our dataset includes **summaries distilled from the top-k responses**, an artifact explicitly designed for thought-level fusion, not available in any prior work.

---

> > ### Author Response · Authors · 2025-12-03
> > **Response to Reviewer 918D (2/2)**
> >
> > # 3. **“No valuable insights or principles on how to design better fusion methods.”**
> >
> > We thank the reviewer for this concern and provide clarification.
> >
> > ### **We believe the paper *does* provide actionable insights for future fusion designs**, specifically:
> >
> > ### **(A) Fusion at different stages produces fundamentally different behaviors**
> > Our comparative results show:
> >
> > - Thought-level fusion is most effective for **reasoning-heavy** tasks (math, code).
> > - Query-level fusion is best for **efficiency-constrained** settings.
> > - Model-level fusion is surprisingly weak for **code** due to training heterogeneity.
> >
> > These observations provide **principled guidance** for the community:
> >
> > > *Fusion should match reasoning intensity and serving constraints of target tasks,*
> > > a principle not articulated in prior single-method fusion papers.
> >
> > ### **(B) Thought-template retrieval is more robust than naive few-shot prompting**
> > We show that **distilled templates** outperform:
> >
> > - direct raw CoT retrieval
> > - standard few-shot prompting
> > - zero-shot prompting
> >
> > This suggests a new design principle:
> >
> > > *Fusion should focus on abstracted reasoning structures rather than full-chain retrieval.*
> >
> > ### **(C) Fusion is harmful for certain domains**
> > Our results reveal that fusion **reduces** performance on:
> >
> > - World Knowledge
> > - Math (model-level)
> >
> > This contradicts the assumption that “fusion always helps,” and produces actionable guidance:
> >
> > > *Fusion must be domain-aware; applying fusion blindly can degrade performance.*
> >
> > ### **(D) LLM-judge scoring is a better selection signal than task metrics**
> > We show that LLM-judge–based selection predicts better training examples for model-level fusion (Table 5).
> > This yields a concrete methodological suggestion:
> >
> > > *Use LLM-judged response quality as a soft selection signal in multi-LLM fusion pipelines.*
> >
> > These principles directly inform how future fusion algorithms should select responses, templates, and fusion stages.
> >
> > ---
> >
> > # Closing Remarks
> >
> > We appreciate the reviewer’s careful analysis.
> > While FusionFactory intentionally does not claim a new “single fusion algorithm,” our contributions lie in:
> >
> > - **introducing the first multi-stage fusion benchmark**,
> > - **revealing cross-level insights previously unknown**, and
> > - **establishing principles that guide future fusion design**.
> >
> > We will revise the paper to more explicitly highlight these contributions in the introduction and discussion sections.
> >
> > Thank you again for your time and constructive feedback.

---

### Note · Authors · 2026-01-05

I have read and agree with the venue's withdrawal policy on behalf of myself and my co-authors.